# Grounding Functional Similarity by Invariance-Aware Model Stitching

Ioannis Athanasiadis [1]   Anmar Karmush [1]   Michael Felsberg [1]

## Abstract

In deep learning, functional similarity evaluation quantifies the extent to which independently trained models learn similar input–output relationships. In model stitching, functional similarity is framed as representation forward compatibility, i.e., whether the representations of two models can be aligned to solve a given task. Recent studies, however, highlight a critical limitation: models relying on different information cues can still produce compatible representations, making them appear misleadingly similar (Smith et al., 2025). We attribute this failure to standard model stitching being inherently blind to the invariance properties of the stitched models. To address this limitation, we introduce the forward–backward compatibility requirement under which we formulate the invariance-aware model stitching. Through analyzing key stitching configurations, we study the interplay between forward and backward compatibility, showing that invariance-aware model stitching provides a more principled approach to functional similarity evaluation while revealing functional discrepancies previously obscured.

## 1. Introduction

Deep neural networks lie at the forefront of modern AI, driving significant advances across various domains ranging from computer vision (Krizhevsky et al., 2017) and natural language processing (Mikolov et al., 2013) to healthcare (Esteva et al., 2017). Their success is attributed to their ability to learn meaningful representations of the data (Bengio et al., 2013; Chowers & Weiss, 2023) which capture their abstract relationships (Doimo et al., 2020; Ziyin et al., 2025). Understanding the emergence of these representations constitutes an important problem (Kornblith et al., 2019) from both scientific and applied perspectives (Ding et al., 2021).

[1]Department of Electrical Engineering, Linköping University, Sweden. Correspondence to: Ioannis Athanasiadis <ioannis.athanasiadis@liu.se>.

*Proceedings of the 43rd International Conference on Machine Learning*, Seoul, South Korea. PMLR 306, 2026. Copyright 2026 by the author(s).

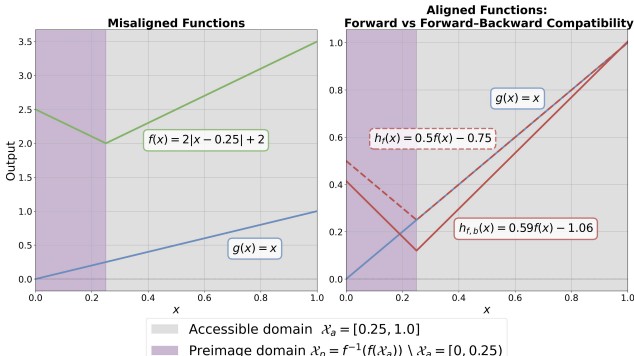

*Figure 1.* A toy numerical example on function alignment under L2 minimization. When two functions $f(x)$ and $g(x)$ are aligned through an affine transformation $h(x) = W \cdot f(x) + b$ only on the accessible domain $\mathcal{X}_a$ (forward compatibility), they appear perfectly aligned (see $h_f(x)$). However, accounting for the preimage domain $\mathcal{X}_p$ (forward–backward compatibility) reveals functional misalignment even within the accessible domain (see $h_{f,b}(x)$). Note that the accessible domain and preimage domain are linked through function $f$.

One approach to gaining insight into learned representations is to compare the similarity of internal representations of different models. In the literature, such similarities generally fall into two categories: *representational* or *functional* similarity (Csiszárik et al., 2021). We refer to Klabunde et al. (2025) for a comprehensive survey of commonly used representational and functional similarity metrics. In this study, we focus on the latter and formulate a functional similarity metric that quantifies the degree to which independently trained neural networks share similar bidirectional input–output relationships.

Representational similarity metrics measure correlation between geometric structures in the embedding spaces of the networks (Raghu et al., 2017; Morcos et al., 2018; Kornblith et al., 2019). Although widely used (Mirzadeh et al., 2021; Nguyen et al., 2021; Masarczyk et al., 2023; Ciernik et al., 2025), these metrics have been criticized for overestimating similarity due to spurious feature correlations (Jones et al., 2022), for being difficult to interpret (Bansal et al., 2021), and for being agnostic to functional behavior (Bansal et al., 2021; Davari et al., 2023; Bo et al., 2025) and invariance properties (Nanda et al., 2022) of the networks.

On the contrary, functional similarity metrics focus on the

input–output behavior of the networks. Depending on the exact definition, different notions of similarity are captured. For example, the hard (Geirhos et al., 2020) and soft (Goel et al., 2025) *error consistency* measure the extent to which identical inputs are assigned to similar outputs by different networks. On the other hand, model stitching (Bansal et al., 2021; Csiszárik et al., 2021) formulates functional similarity as embedding compatibility, i.e., whether the representations of one model can be used by another one such that a functional property of interest (e.g., classification accuracy) is maintained. In practice, this is realized by dividing each network into two parts, where the first part of one network, the *front model*, is plugged into the second part of another, the *end model*, through a trainable affine transformation that aligns the two. The similarity is then defined as the functional property achieved by the stitched composition.

Notably, model stitching has been used to demonstrate that independently trained networks (i.e., different initialization, training objectives etc.) share compatible internals (Bansal et al., 2021; Csiszárik et al., 2021; Balogh & Jelasity, 2023). Recent works, however, questioned the reliability of this conclusion under the commonly used stitching setting, task loss matching (Hernandez et al., 2022; Balogh & Jelasity, 2025; Smith et al., 2025). For example, Smith et al. (2025) demonstrated stitching compatibility between networks trained to utilize different visual cues, when solving the same task, and argued that high similarity in these cases is "misleading", a view we also share. In line with this perspective, we argue that a *meaningful* notion of functional similarity should reflect the extent to which two models rely on similar information cues (i.e., similar input patterns) when solving their respective tasks.

Standard model stitching quantifies the impact of interchanging the stitch-level representations on some functional property at the output level (i.e., forward direction), effectively probing for *forward compatibility*. Different formulations of forward compatibility are realized depending on the objectives used to establish the stitching alignment. For example, in model stitching under task loss matching, **output-level forward compatibility** is interpreted as similarity. In the other extreme, when minimizing the representation distance only at the stitch level, that is model stitching under direct matching (DM), **stitch-level forward incompatibility** is interpreted as dissimilarity. Ultimately, relying solely on forward compatibility can obscure functional discrepancies in the preimage of stitch-level representations (i.e., backward direction), leading to spurious assessments of functional similarity. Conversely, probing only for *backward compatibility* [1] measures shared invariances (Nanda et al., 2022; Feather et al., 2023) between the two models, but not whether their representations are (functionally) interchangeable (i.e., forward compatible). For example, two models may share invariances without being forward compatible, and vice versa. In this regard, accounting for both forward and backward compatibility notions emerges as a relevant avenue for functional similarity evaluation. We conceptually motivate our view through a toy functional alignment example shown in Fig. 1.

Towards model stitching that better fits the needs of functional similarity evaluation, we strive for developing a setting that jointly accounts for functional nuances in both directions. To this end, we pose the *forward–backward compatibility* requirement under which the stitched models are considered functionally similar.

**Requirement. Forward–backward compatibility:** For each pair of inputs that the front model represents identically at the stitch level, the standalone end model, and the end model within the stitched composition process the stitch-level representation in a similar manner [2].

In practice, we probe for forward–backward compatibility by using *identically represented inputs* (IRIs) (Nanda et al., 2022) to establish stitching alignment over the invariance classes induced by the front model at the stitch level. Model stitching under the forward–backward compatibility notion is characterized as *invariance-aware*, since the resulting composition depends on how the invariances induced by the front model are interpreted by the end model. Pairing forward and backward compatibility naturally requires that the former incorporates supervision from the standalone end model when establishing the alignment. We refer to these formulations as *invariance-enabling*. Formally, our contributions are as follows:

- We show that invariance-aware model stitching mitigates previously reported failure modes of standard model stitching (Smith et al., 2025), yielding a more principled approach to functional similarity evaluation (see Secs. 3.1 and 3.2).

- In light of the above, we revisit model stitching between robust and non-robust networks revealing that they are not as functionally similar as suggested by the literature (see Sec. 3.3).

- Towards understanding the interplay between forward and backward compatibility, we analyze key stitching settings. Our analysis leads to two main observations: (i) forward–backward compatibility is applicable to all invariance-enabling formulations, yielding qualita-

---

[1]The term *backward* refers to the direction of information flow and is not related to gradient backpropagation.

[2]The notion of similarity depends on the exact formulation of forward compatibility (e.g., output-level forward–backward compatibility under the task loss matching objective).

tively analogous conclusions (see Secs. 3.1, 3.2 and 3.3), and (ii) the level of supervision induced by the exact forward compatibility formulation modulates a trade-off between establishing non-trivial functional alignment and avoiding reliance on previously unseen cues (see Sec. 3.4).

The code is available at https://github.com/athaioan/invariance-aware-stitching.

## 2. Model Stitching

In this section, we provide an overview of the components forming the basis of our experimental setup. Specifically, we introduce the model stitching notation (Sec. 2.1), we define the training objectives used to establish the stitching alignment (Sec. 2.2) and describe the targeted training data manipulations relevant to our study (Sec. 2.3).

### 2.1. Notation

We closely follow the notation used by Balogh & Jelasity (2025). Let $f_w : \mathcal{X} \to \mathcal{Y}$ be a feedforward neural network with $m$ layers, parameterized by $w$, which we omit for brevity when clear from the context. We can write $f$ as the composition $f := f_m \circ \cdots \circ f_1$, where each $f_i : \mathcal{A}_{i-1} \to \mathcal{A}_i$ maps the activation space of the $(i-1)^{\text{th}}$ to that of the $i^{\text{th}}$, with $\mathcal{A}_0 = \mathcal{X}$. For $i \geq 1$ and $i \leq r \leq m$, we define the partial compositions $f_{[i:r]} := f_r \circ \cdots \circ f_i$ as well as $f_{>i} := f_{[i+1:m]}$ and $f_{\leq i} := f_{[1:i]}$ such that $f = f_{>i} \circ f_{\leq i}$. Similarly, we assume $g_\phi : \mathcal{X} \to \mathcal{W}$ with $g := g_k \circ \cdots \circ g_1$ and $g_j : \mathcal{Z}_{j-1} \to \mathcal{Z}_j$.

Model stitching for functional similarity evaluation aims at quantifying the extent to which $g_{>j}$ can maintain the functionality of $g$ when given as input the $\mathcal{A}_i$ with $i \in [m]$ and $j \in [k]$. Formally, the stitched model is constructed as $h_\theta = g_{>j} \circ T_\theta \circ f_{\leq i}$. Here, $T_\theta : \mathcal{A}_i \to \mathcal{Z}_j$ is referred to as the stitching layer, and we refer to $f$ and $g$ as front and end models respectively. In practice, the optimal $T_\theta$, under a relevant optimality criterion, is selected from a suitable family of transformations $\mathcal{S}$, while $g$ and $f$ are kept fixed. Given the nature of the problem, the transformation $T_\theta \in \mathcal{S}$ needs to be sufficiently flexible to allow non-trivial mappings while ensuring that the capacity of $h$ does not exceed that of the combined partial networks $g_{>j}$ and $f_{\leq i}$. The family $\mathcal{S}$ of affine transformations is considered in the literature to satisfy these requirements (Balogh & Jelasity, 2025).

In our work, we considered functional similarity evaluation of image classification networks, the predominant use case in the relevant literature. We assume access to two classification datasets, $\mathcal{D}_{\text{train}} = \{(x_s, y_s)\}_{s=1}^n$ and $\mathcal{D}_{\text{test}} = \{(x_s, y_s)\}_{s=1}^c$, where $x_s$ denotes an image and $y_s$

its corresponding label in one-hot vector format. The transformation $T_\theta$ is optimized using $\mathcal{D}_{\text{train}}$, while the functional similarity is evaluated on the $\mathcal{D}_{\text{test}}$.

### 2.2. Stitching Alignment Objectives

The optimization objectives previously explored in the literature are the hard label matching (HLM) (Bansal et al., 2021; Csiszárik et al., 2021), the soft label matching (SLM) (Csiszárik et al., 2021) [3] and the direct matching (DM) (Csiszárik et al., 2021; Lähner & Moeller, 2024; Balogh & Jelasity, 2025) defined as:

$$\mathcal{L}_{\text{HLM}} : \arg\min_\theta \mathbb{E}_{p(x,y)}\Big[\mathcal{L}_{\text{CE}}\big(h_\theta(x), y\big)\Big], \quad (1)$$

$$\mathcal{L}_{\text{SLM}} : \arg\min_\theta \mathbb{E}_{p(x)}\Big[\mathcal{L}_{\text{CE}}\big(h_\theta(x), g(x)\big)\Big], \quad (2)$$

$$\mathcal{L}_{\text{DM}} : \arg\min_\theta \mathbb{E}_{p(x)}\Big[\big\|(T_\theta \circ f_{\leq i})(x) - g_{\leq j}(x)\big\|_F\Big], \quad (3)$$

with $\mathcal{L}_{\text{CE}} : \mathcal{Y} \times \mathcal{Y} \to \mathbb{R}$ denoting the cross-entropy loss.

When it comes to forward compatibility, HLM and SLM may be overly **forgiving**, as they can exploit irregularities in the models' decision processes to achieve optimal output-level matching performance. Conversely, DM can be overly **penalizing**, since by construction, stitch-level matching does not account for the subspaces relevant to the layers (i.e., the flow of activations) following the stitching.

Inspired by the feature distillation literature (Romero et al., 2015; Liu et al., 2023) and towards understanding the connection between the output- and stitch-level matching, we propose a novel latent-level stitching formulation termed functional latent alignment (FuLA). Under FuLA, the stitched composition is explicitly optimized to mimic the internal processes from the stitching layer up to the penultimate layer of the standalone end model. Formally, the $\mathcal{L}_{\text{FuLA}}$ objective is defined as:

$$\arg\min_\theta \mathbb{E}_{p(x)}\Bigg[C_j \underbrace{\frac{\big\|(T_\theta \circ f_{\leq i})(x) - g_{\leq j}(x)\big\|_F}{\|g_{\leq j}(x)\|_F}}_{\mathcal{L}_{\text{Hint}}^j}$$
$$+ \sum_{l=j+1}^{k-1} C_l \underbrace{\frac{\big\|g_{[j+1:l]}\big((T_\theta \circ f_{\leq i})(x)\big) - g_{\leq l}(x)\big\|_F}{\|g_{\leq l}(x)\|_F}}_{\mathcal{L}_{\text{Hint}}^l}\Bigg], \quad (4)$$

---

[3] Csiszárik et al. (2021) used the task loss matching term to refer to both HLM and SLM interchangeably and argued that these two formulations yield highly correlated results. However, we find that HLM and SLM can behave differently under both standard and, in particular, invariance-aware model stitching, therefore, we distinguish between them.

where $C \in \{c \in \mathbb{R}^{k-j} : c_l \geq 0, \sum_{l=j}^{k-1} c_l = 1\}$ defines a weighted average that controls the relative contribution of each term. We use uniform weighting as the default option. Additionally, we divide by the target's norm to account for potential differences in scale between feature activations at different depths. We refer to these terms as Hints (Romero et al., 2015), denoted as $\mathcal{L}_{\text{Hint}}^t$, where $t$ indicates the depth. For a visual overview of the model stitching objectives, we refer readers to Fig. 6 in the appendix.

## 2.3. Training Data in Model Stitching

In this work, we leverage the concepts of identically represented inputs (IRIs) (Nanda et al., 2022) and *adversarial training (AT)* (Madry et al., 2018) both of which involve altering the training data used to establish the alignment in model stitching.

### 2.3.1. GENERATING IDENTICALLY REPRESENTED INPUTS (IRIS)

Given an input $x$ and a feature extractor $f_{\leq i}$, for a small enough tolerance $\rho > 0$ we define the set:

$$\text{IRIs}_r(x; f_{\leq i}) = \{ \, x' \mid \underbrace{\frac{||f_{\leq i}(x') - f_{\leq i}(x)||_F}{||f_{\leq i}(x)||_F}}_{\mathcal{L}_{\text{Hint}}^i} \leq \rho \}, \quad (5)$$

which consists of all inputs $x'$ that yield *almost* the same representation as $x$ under $f_{\leq i}$. For $\rho = 0$, this reduces to the exact representation-invariant set $\text{IRIs}(x; f_{\leq i})$, which is hard to attain in practice. Following Nanda et al. (2022), we obtain $\text{IRIs}_r(x; f_{\leq i})$ by optimizing trainable inputs $x'$, initialized by uniform random noise, to minimize the IRIs Hint $\mathcal{L}_{\text{Hint}}^i$. We then construct the dataset $\mathcal{D}_{\text{train}}^{\text{IRIs}} = \{(x'_s, y_s)\}_{s=1}^n$ by replacing each input $x_s$ from $\mathcal{D}_{\text{train}}$ with a corresponding sample $x'_s \in \text{IRIs}_r(x_s; f_{\leq i})$, while we consider $f$ as the front model which we stitch into the end model $g$ at stitch level $i$.

By construction, corresponding pairs between $\mathcal{D}_{\text{train}}^{\text{IRIs}}$ and $\mathcal{D}_{\text{train}}$ are *almost* indistinguishable to $f$ at the stitch level. We assume that optimizing the stitching layer on $\mathcal{D}_{\text{train}}$ achieves the *unique* optimal forward compatibility on $\mathcal{D}_{\text{test}}$. Under this assumption, any deviation from this optimum, observed when training on $\mathcal{D}_{\text{train}}^{\text{IRIs}}$, constitutes a violation of the forward–backward compatibility requirement. In practice, we probe this requirement by training the stitching alignment on $\mathcal{D}_{\text{train}}^{\text{IRIs}}$ while evaluating the functional similarity on $\mathcal{D}_{\text{test}}$. We refer to this operationalization as invariance-aware model stitching.

A methodology similar in nature was employed by Jones et al. (2022), where CKA was computed on a representation-inverted dataset, effectively realizing an invariance-aware representational similarity metric that nonetheless inherits the main limitations of the metrics in this category (Bansal et al., 2021) (e.g., agnostic to functional behavior and lacking interpretability). In contrast, invariance-aware model stitching constitutes a functional similarity metric that quantifies the degree of representation interchangeability while also being sensitive to non-shared invariances, thereby jointly capturing (dis)similarity in the forward and backward directions.

### 2.3.2. ADVERSARIAL TRAINING (AT)

AT is widely regarded as the most effective method for improving model robustness against worst-case perturbations, referred to as adversarial examples (Goodfellow et al., 2015; Li et al., 2022). In this case, the training objective for the neural network $f_w$ becomes:

$$
\begin{aligned}
\arg \min_w \, & \mathbb{E}_{p(x,y)} \big[ (1 - \alpha) \mathcal{L}_{\text{CE}}(f_w(x), y) \\
& + \alpha \max_{\delta \in \mathcal{B}(x,\epsilon)} \mathcal{L}_{\text{CE}}(f_w(x + \delta), y) \big],
\end{aligned} \quad (6)
$$

where $\alpha \in [0, 1]$ controls the proportion of adversarial samples and $\mathcal{B}(x, \epsilon)$ denotes the $\ell_\infty$-ball of radius $\epsilon$. In this work, we revisit the functional similarity between robust and non-robust networks (Balogh & Jelasity, 2023) where we train robust models via AT to serve as the front and/or end models. Additionally, we leverage AT during the optimization of the stitching composition $h_\theta$ (i.e., replacing $f_w$ with $h_\theta$ in Eq. (6)) to evaluate functional alignment with respect to adversarial samples.

## 3. Analyzing Functional Similarity via Model Stitching

We posit that relying solely on forward compatibility is insufficient for meaningful functional similarity evaluation in model stitching, and that probing for forward–backward compatibility provides a more principled setting in this context. Towards understanding the interplay between the different formulations of forward compatibility (e.g., stitch- or latent-level) and backward compatibility, we compare the functional similarity achieved for both standard and invariance-aware model settings for all stitching objectives presented in Sec. 2.2.

Note that hard label matching (HLM) operates exclusively on the front model and the hard labels, therefore is not invariance-enabling. Based on that, probing for forward–backward compatibility under HLM serves as a proxy for the semantic consistency of the $\mathcal{D}_{\text{train}}^{\text{IRIs}}$, since regular data and its corresponding exact identically represented inputs (IRIs) would be indistinguishable under HLM. In the absence of

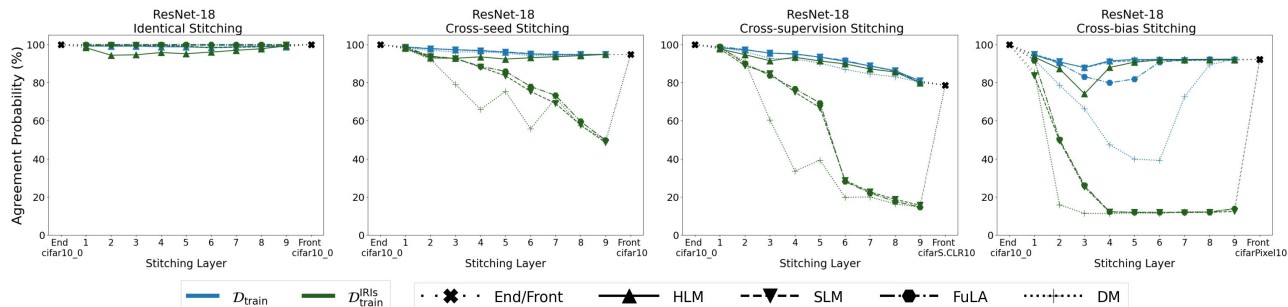

*Figure 2.* Stitching between models relying on different information. Probing for forward–backward compatibility leads to significantly different, and fundamentally more intuitive functional similarity trajectories compared to only probing for forward compatibility.

objective measures of meaningfulness in this context, we rely on sanity checks grounded on intuitive expectations (see Secs. 3.1 and 3.2), which have formed the basis of critiques in recent literature.

**Stitching layer:** The stitching layer $T_\theta$ is modeled as a $1 \times 1$ convolutional layer including a bias term, while keeping the front and end models frozen, following Csiszárik et al. (2021). We initialize $T_\theta$ by linearizing the direct matching (DM) objective in Eq. (3) and solving it using the Moore-Penrose pseudoinverse with 500 training samples (Csiszárik et al., 2021; Balogh & Jelasity, 2025) from $\mathcal{D}_{\text{train}}$. According to Csiszárik et al. (2021), the initialization scheme affects the final alignment following stitching optimization. Based on that, we used the same initialization scheme for both standard and invariance-aware model stitching to ensure that observed differences are not driven by initialization.

**Experimental setting:** We consider image classification networks and conduct our experiments using CIFAR-10 (Krizhevsky, 2009) and LS-ImageNet-10, a low-resolution version of ImageNet (Deng et al., 2009) from which we randomly sampled 10 classes, and modified variants of these (e.g., variants containing shortcuts). Throughout the paper, we use the ResNet-18 (He et al., 2016) architecture as the trained front and end models. We perform stitching at the first convolutional layer and at each residual block. For each stitching configuration, we report the functional similarity averaged across three random initializations.

**The stitching plot:** When stitching corresponding layers (i.e., $i = j$), we use the stitching plot (Balogh & Jelasity, 2023), where the y-axis represents a relevant functional property and the x-axis denotes the depth of the front layer composition. The first and last points in the x-axis correspond to the baseline end and front models respectively. Previous works use top-1 accuracy of the stitched composition on the end model's task to infer functional similarity. However, we argue that top-1 measures can overlook finer aspects of similarity (e.g., per-sample predictions). Instead, we utilize the *agreement probability* (Goel et al., 2025) with the end model's predictions as described in App. C, captur-

ing similarity of the full class probability distributions in a sample-wise manner.

By design, the stitching plot between a front and an end model at $M$ corresponding layers begins at $(0, 100\%)$ and ends at $(M + 1, Q)$, where $Q$ is the agreement probability achieved by the standalone front model relative to the end model. For example, a data point at $(3, 95\%)$ suggests that replacing the first three layers of the end model with those of the front results in a stitched composition with $95\%$ agreement probability, with respect to the standalone end model's predictions. Intuitively, we expect the stitching plots of similar networks to resemble a smooth transition from $100\%$ to $Q$ agreement probability.

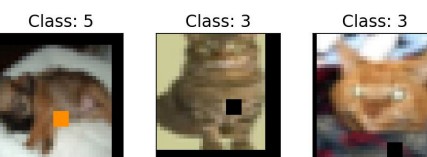

*Figure 3.* Pixel shortcuts. Note that the color of the injected pixel is predictive of the image class (e.g., ■ for "cat").

### 3.1. Functional Similarity across Information Variants

Smith et al. (2025) showed that models trained to rely on different cues (e.g., color, bias, etc.) are functionally similar under HLM, and argued that this is an undesirable property of functional similarity evaluation. We revisit this observation by stitching front models trained on different CIFAR-10 variants into **the same** end model trained on CIFAR-10. In this setup, any deviation from the perfect agreement curve can be attributed to incompatibility originating from the front model.

In particular, we stitch into the end model the following front models, forming a spectrum of increasing semantic divergence: (1) the end model itself (identity), (2) a different random initialization of the end model trained on CIFAR-10 (cross-seed), (3) a model trained on CIFAR-10 under

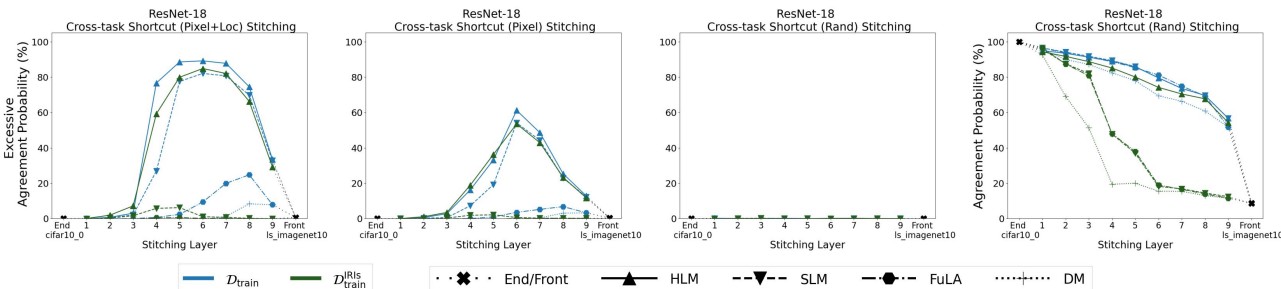

*Figure 4.* Stitching under shortcuts of varying availability. When probing solely for forward compatibility, all its formulations are susceptible to exploiting previously unseen yet predictive cues to establish alignment, as indicated by elevated excessive agreement probability. Probing for forward–backward compatibility substantially mitigates this behavior.

the SimCLR (Chen et al., 2020) self-supervised objective followed by fully supervised tuning of its classifier (cross-supervised), and (4) a model trained on CIFAR-10 with injected class-correlated pixel shortcuts (cross-bias) (see Fig. 3). For configurations (1)–(3), we optimize the stitching layer and report similarity on the CIFAR-10 train ($\mathcal{D}_{\text{train}}$) and test ($\mathcal{D}_{\text{test}}$) splits respectively. For configuration (4), we use the pixel-injected versions of these splits to ensure that both the front and end models have access to their predictive visual cues.[4]

Fig. 2 shows that under standard model stitching (blue curves), all formulations achieve high functional similarity in the deep layers, even in the cross-bias case. This suggests that the limitation identified by Smith et al. (2025) is not specific to the HLM formulation, but instead appears to arise from exclusively probing for forward compatibility. In contrast, under the invariance-aware settings (green curves, excluding HLM), in accordance with our intuition, the functional similarity in the cross-bias configuration quickly diminishes to random chance (i.e., non-existent functional similarity). Another key difference between standard and invariance-aware settings is that only under the latter we observe a progressively sharper decrease in functional similarity as we transition from (1)-(4), reflecting the increasing semantic divergence between the front models and the end model. Based on these, we conclude that invariance-aware model stitching provides a more meaningful notion of functional similarity compared to standard model stitching. Among the invariance-enabling formulations, functional latent alignment (FuLA) and soft label matching (SLM) achieved comparable functional similarity while DM achieved consistently lower similarity, indicating that the DM misalignment emerges, for the most part, already at the latent level (i.e., the feature representations following the stitch level).

---

[4]Models trained on pixel-injected CIFAR-10 perform only marginally above random chance on CIFAR-10, whereas models trained on standard CIFAR-10 perform comparably on both standard and pixel-injected versions.

Finally, the HLM formulation under invariance-aware model stitching behaves similarly to those of the standard model stitching, indicating that the relaxed IRIs$_\text{r}$ set provides a good approximation of the exact IRIs set while preserving its semantic structure. Consequently, the observed behavior cannot be attributed to poorly constructed $\mathcal{D}_{\text{train}}^{\text{IRIs}}$, a conclusion further supported by additional sanity checks (see App. F.1).

## 3.2. Functional Similarity under Unseen Predictive Information

In Sec. 3.1, we revisited model stitching between models relying on different information cues in cases where both models have access to their corresponding **previously seen** predictive cues. We showed that invariance-aware model stitching can distinguish between such models and therefore is better positioned for meaningful functional similarity evaluation compared to standard model stitching. A related, yet distinct failure mode of standard model stitching was also surfaced by Smith et al. (2025). Namely, that of the stitching composition opportunistically exploiting information cues that are predictive of the alignment task but were **previously unseen** by the stitched models, a property highly undesirable in the context of functional similarity evaluation (Bansal et al., 2021). A natural next step is to evaluate how invariance-aware compares to standard model stitching in this regard. To address this we perform our analysis across all stitching objectives used earlier while also controlling for unseen cue accessibility.

To this end, we introduce an additional front model: (5) a model trained on LS-ImageNet-10, which is stitched into the same CIFAR-10 end model used earlier (cross-task). To establish and evaluate the alignment, we consider three variants of CIFAR-10: (i) pixel-injected data at a fixed location for each class, (ii) pixel-injected data, and (iii) data from (ii) where the pixel patterns are replaced with random noise (i.e., negating the shortcuts). Among these, (i) provides the most available and predictive cues followed by (ii), whereas (iii) provides none. Note that neither the front nor the end

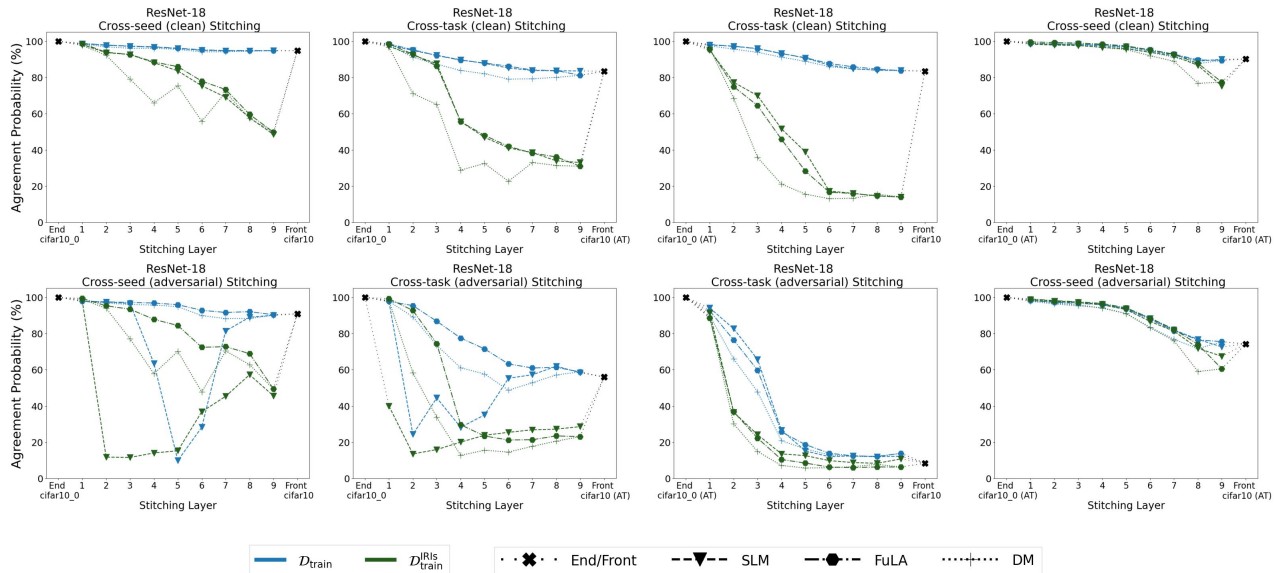

*Figure 5.* Stitching between robust and non-robust models. The "clean" and "adversarial" labels indicate that the stitching layer is optimized and evaluated using only benign or adversarial samples respectively. Probing for forward–backward compatibility reveals functional incompatibilities previously overlooked. For example, probing only for forward compatibility indicates that the robust and non-robust models are functionally similar with respect to benign samples (i.e., first row), with functional incompatibilities revealed upon probing for forward–backward compatibility.

models have previously encountered these class-correlated patterns, therefore any utilization of those originates from the stitching layer exploiting them.

To quantify the extent to which the stitched composition exploits the pixel patterns, Fig. 4 reports the excessive agreement probability, defined as the difference between the agreement probability on the pixel-injected test split and that on a variant in which pixel patterns are permuted across classes. Intuitively, a larger difference indicates a greater reliance on the pixel cues presented during training.

Under the standard model stitching setting, we observe that all formulations are susceptible to learning shortcuts, with the effect diminishing as we progress from (i) to (iii). These results indicate that the effect of exploiting unseen cues is not exclusive to the HLM formulation but is a property of forward compatibility. This finding casts further doubt on forward compatibility as a proxy for functional similarity evaluation.

Fortunately, invariance-aware model stitching substantially mitigates this shortcoming. In particular, we observe that, across all invariance-enabling formulations, excessive agreement is greatly reduced, often approaching zero, closely matching the behavior observed in configuration (iii). When comparing similarity in configuration (iii), we find that DM achieves the lowest functional similarity, while FuLA and SLM remain comparable. As argued earlier, this suggests that misalignment in DM arises already at the representation level immediately following the stitching layer.

## 3.3. Functional Similarity between Robust and Non-Robust Models

Perhaps surprisingly, Balogh & Jelasity (2023) showed that robust and non-robust networks are functionally similar under model stitching by HLM. As we saw earlier, forward compatibility not only fails to distinguish between models that rely on different cues to solve a task, but can also exploit cues unintended by either the front or end models. On the other hand, forward–backward compatibility, as operationalized by invariance-aware model stitching, was found to significantly mitigate these shortcomings. These findings motivate revisiting the question of functional similarity between robust and non-robust networks under the forward–backward compatibility notion. Additionally, the multi-objective nature of the robust-to-non-robust configurations makes it a relevant test bed for evaluating the interplay between stitch-, latent- and output-level forward compatibility formulations.

We consider four models: the two non-robust models (1) and (2) used earlier, both trained on CIFAR-10 from different random initializations, along with (6) a robust model trained on CIFAR-10 using AT of Eq. (6) with $\alpha = 0.5$ and (7) a different random initialization of (6). Similar to Balogh & Jelasity (2023), we use these models to construct all possible robust/non-robust stitching configurations. In this case, the similarity with respect to both clean (i.e., benign) and adversarial samples is of interest. To avoid conflating the similarity associated with different sample types, we op-

timize the stitching layer and evaluate similarity on clean and adversarial samples independently, thereby evaluating functional similarity with respect to each sample type in isolation. As established earlier, HLM is a suboptimal formulation in this context, as it is highly prone to undesirable shortcut exploitation under model stitching. For this reason, and for presentation brevity, we omit HLM from the remainder of the analysis, using SLM as the sole representative of output-level compatibility formulations.

In Fig. 5 we provide the results obtained when stitching between various combinations of robust and non-robust models. Under standard model stitching, the DM and FuLA formulations display a smooth similarity transition during stitching, indicating compatibility for both clean and adversarial samples, for all configurations. On the other hand, SLM degenerates in cases involving non-robust end models where the stitching alignment was established on adversarial examples (see Fig. 5, bottom row, left two panels). Under the invariance-aware settings, functional discrepancies between robust and non-robust models become visible allowing for more fine-grained similarity evaluations. Interestingly, under invariance-aware model stitching functional similarity is consistently higher between robust models compared to all other configurations. This observation also supports the view that robust models converge to a shared representation structure as argued by Jones et al. (2022). Analogous observations were made by Nanda et al. (2022) and Feather et al. (2023) where non-robust models were shown to develop idiosyncratic invariances that are alleviated with adversarial training.

Finally, under both the standard and invariance-aware model stitching, DM generally achieved the lowest similarity with the only exception being the two SLM degenerated cases discussed earlier while FuLA and SLM performed comparably in all other cases.

### 3.4. Further Results and Discussion

We repeated the experiments on the VGG-16 (Simonyan & Zisserman, 2015) and ViT-Tiny (Dosovitskiy et al., 2021) architectures and further extended the analysis to a higher-resolution dataset (ImageNet-100 (Sarıyıldız et al., 2023)) using ResNet-18 and ViT-Base (Dosovitskiy et al., 2021).

**In relation to Sec. 3.1:** the claims generalize (see App. F.2), as all forward compatibility formulations are susceptible to reporting high functional similarity in cross-bias configurations, at least at the penultimate layer. Moreover, the results further support the relevance of invariance-aware model stitching, which yields a progressively sharper decrease in functional similarity as the semantic divergence of the front model increases.

**In relation to Sec. 3.2:** the claims generalize (see App. F.3),

where in all stitching configurations, at least output-level (and in most cases all) forward compatibility formulations are susceptible to exploiting previously unseen cues to improve alignment. Invariance-aware model stitching consistently mitigates this effect across all invariance-enabling formulations.

**In relation to Sec. 3.3:** the claims made generalize (see App. F.4), where invariance-aware model stitching reveals functional discrepancies previously obscured by standard model stitching performed between robust and non-robust models. Moreover, in all cases, we observed that under invariance-aware model stitching robust models functionally converge to a greater extent than non-robust models.

**Identifying the optimal invariance-enabling forward compatibility:** We have established that invariance-aware model stitching constitutes a more principled approach to functional similarity evaluation. However, out of all available stitching formulations, which is the optimal? In this regard, there is no clear winner, as all formulations displayed favorable properties for at least one architecture. For ViT-Tiny, stitch-level forward–backward compatibility was the formulation that eliminated the unseen cue exploitation most effectively (see Figs. 12 and 14 in the appendix). Latent-level forward–backward compatibility matched its stitch-level counterpart in eliminating unseen cue exploitation in all but ViT-Tiny architectures, with FuLA generally achieving higher functional similarity than DM in all cases. On the other hand, the output-level forward–backward compatibility formulation, as realized by SLM, generally achieves the highest functional similarity in transformer-based architectures at the expense of a higher degree of unseen cue exploitation (see App. F.3). Overall, our experiments reveal a trade-off between achieving non-trivial alignments with high functional similarity and avoiding reliance on previously unseen cues.

**Connection to representation similarity:** Finally, we compared FuLA as a representative model stitching formulation with CKA, as well as contrasted both with their invariance-aware analogues (i.e., by using IRIs (Nanda et al., 2022)), where we conclude that invariance-aware model stitching offers unique insights on model similarity (see App. G).

## 4. Related Work

**Model Stitching:** Model stitching (Lenc & Vedaldi, 2015) has been used to connect independent network components from the model zoo (Yang et al., 2022; Pan et al., 2023; 2024), allowing for networks with controllable performance-efficiency trade-offs. Moreover, it was shown that models can be successfully stitched either by direct transformation in the latent space (Maiorca et al., 2023; Lähner & Moeller, 2024) or in zero-shot fashion by learning representations

in relative space (Cannistraci et al., 2023; Moschella et al., 2023; Cannistraci et al., 2024). In contrast to these works, we employ model stitching as a means for functional similarity evaluation (Bansal et al., 2021; Csiszárik et al., 2021), where task performance is required to correlate with similarity.

**Feature distillation:** Feature-based distillation was introduced by Romero et al. (2015), where in principle, student features are trained to mimic the appearance of corresponding teacher features via L2 minimization. More recently, Liu et al. (2023) formalized the notion of function-consistent feature distillation under which student features replicate the functionality of the teacher's features. Other notable extensions include attention-based distillation (Zagoruyko & Komodakis, 2017) and inter-channel correlation transfer (Liu et al., 2021), both applied at manually chosen layers. Ji et al. (2021) and Chen et al. (2021) avoid manual layer linking by learning connections via attention. Although our work also involves feature distillation at different levels, we perform it exclusively through frozen layer compositions, while considering features activated by metamers of regular inputs.

## 5. Conclusion

In this work, we examine the problem of functional similarity evaluation through model stitching. Previous findings (Smith et al., 2025) suggest that the hard label matching (HLM) model stitching formulation cannot reliably distinguish between models relying on different information cues, while it is also susceptible to exploiting predictive, yet previously unseen, information cues to improve alignment. Both of these are undesirable properties in the context of functional similarity evaluation. We further demonstrate that this behavior generalizes to other forward compatibility formulations, including the less explored soft label matching (SLM) and our newly proposed functional latent alignment (FuLA). To mitigate these issues, we propose invariance-aware model stitching, operationalizing the notion of forward–backward compatibility, where functional similarity is probed in a bidirectional manner. Alongside this, we revisit stitching between robust and non-robust models, yielding insights that differ from previous literature, which considered only forward compatibility. Overall, our findings highlight the importance of evaluating similarity from multiple perspectives to obtain a more fine-grained understanding of the inner workings of neural networks.

**Limitations:** An inherent limitation of this work is the absence of an "oracle" notion of functional similarity. As a result, definitive conclusions about which settings provide the most faithful reflection of similarity remain out of reach. Finally, our study is limited to image classification models. Exploring the interplay of forward and backward compatibility in model stitching for other tasks within the same modality and across different modalities is a promising direction for future work.

## Impact Statement

Our work contributes to the field of machine learning by developing frameworks to better understand the inner workings of deep neural networks. By improving our understanding of how neural networks operate internally, we can guide targeted interventions that align with societal needs, such as promoting fairness, inclusivity, and robustness in AI systems.

## Acknowledgements

We thank Pavlo Melnyk for providing constructive feedback on an early draft of the manuscript. This work was supported by the Wallenberg Artificial Intelligence, Autonomous Systems and Software Program (WASP), funded by the Knut and Alice Wallenberg Foundation. The computational resources were provided by the National Academic Infrastructure for Supercomputing in Sweden (NAISS) at C3SE, partially funded by the Swedish Research Council through grant agreement no. 2022-06725.

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

# A. Constructing $\mathcal{D}_{\text{train}}^{\text{IRIs}}$

Given a front model $f$ and stitch level $i$, we construct the $\mathcal{D}_{\text{train}}^{\text{IRIs}}$ by sampling a data point $x'$ from the $\text{IRIs}_{\text{r}}(x; f_{\leq i})$, for each data point x in $\mathcal{D}_{\text{train}}$.

$$\text{IRIs}_{\text{r}}(x; f_{\leq i}) = \{\, x' \mid \underbrace{\frac{||f_{\leq i}(x') - f_{\leq i}(x)||_F}{||f_{\leq i}(x)||_F}}_{\mathcal{L}_{\text{Hint}}^i} \leq \rho \}. \tag{7}$$

In practice, we initialize the $x'$ by drawing from the uniform distribution which we progressively update for 200 steps through minimizing the $\mathcal{L}_{\text{Hint}}^i$. For the low-resolution images, we used a step size of $1/255$ and $\epsilon$ set to 1, whereas for the high-resolution images, we used a step size of $4.758 \times 5/255$ with $\epsilon$ set to $4.758$ (i.e., the scale difference is due to using different data normalization for each setting).

# B. Experimental Setup

In this section, we cover the implementation details needed for reproducing our results.

## B.1. Baseline Training

In this study, we focused on image classification networks and experimented with both convolution- and transformer-based architectures, namely ResNet-18 (He et al., 2016), VGG-16 (Simonyan & Zisserman, 2015), ViT-Tiny and ViT-Base (Dosovitskiy et al., 2021). For the low-resolution settings, we used CIFAR-10 (Krizhevsky, 2009) and a low-resolution version of ImageNet (Deng et al., 2009), from which we randomly sampled 10 classes. We refer to this dataset as LS-ImageNet-10. For the high-resolution setting, we used ImageNet-100 (Sarıyıldız et al., 2023) as well as ImageNetB-100, which we created by randomly sampling a set of 100 classes from ImageNet-1K not included in ImageNet-100.

Note that achieving optimal classification performance across the different standalone models is beyond the scope of this work. Instead, we seek to evaluate functional similarity between independently trained neural networks. In this regard, it is sufficient to ensure that these models train stably and perform reasonably well with respect to their classification tasks, that is, significantly above chance performance.

**Non-robust models:** We considered ResNet-18, VGG-16 and ViT-Tiny for the low-resolution setting and ResNet-18 and ViT-Base for the high-resolution one. We trained all models from scratch closely following the training recipes used by Balogh & Jelasity (2023) while performing minimal tuning when necessary (e.g., for training stability). All models were trained for 200 epochs using the stochastic gradient descent (SGD) optimizer with momentum (Sutskever et al., 2013) with a weight decay penalty set to $1 \times 10^{-4}$, while we also employed standard data augmentation consisting of random horizontal flipping and cropping. The learning rate was initially set to $0.1$ for ResNet-18 and to $0.01$ for the other architectures, where we used a step scheduler that decayed the learning rate by a factor of $0.1$ at $1/3$ and $2/3$ of the training process. We used a batch size of 128 and 256 for the low- and high-resolution settings respectively.

For the self-supervised models, we used identical training hyperparameters, with the only exception being that we used additional random color jittering and grayscale transformations. We initially trained these on the SimCLR (Chen et al., 2020) objective using a temperature of $0.5$ for 200 epochs followed by additional 200 epochs training of the linear classification head while keeping the backbone frozen.

**Robust models:** The robust models were trained using the same hyperparameter settings as the non-robust models, with adversarial training performed according to Madry et al. (2018), using an equal mix of clean and adversarial examples (i.e., $\alpha = 0.5$). In all settings, we used 10 perturbation steps to generate the adversarial examples. For the low-resolution setting, we used a step size of $2/255$ with $\epsilon$ set to $8/255$. For the high-resolution setting, we used a step size of $4.758 \times 1/255$ with $\epsilon$ set to $4.758 \times 4/255$. Following Balogh & Jelasity (2023), we measure adversarial robustness using the "rand" configuration of AutoAttack (Croce & Hein, 2020).

### B.2. Stitching Layer Training

For convolution-based architectures, following Balogh & Jelasity (2023) and Csiszárik et al. (2021), we use $1 \times 1$ convolutional layers with bias as a stitching layer to connect the front and end models. During training, we kept everything frozen apart from the alignment transformation (i.e., the stitching layer). When training on $\mathcal{D}_{\text{train}}$, we allowed the update of the batch normalization statistics of the front and end models, while when training on $\mathcal{D}_{\text{train}}^{\text{IRIs}}$, we also kept the batch statistics frozen to avoid accumulating errors when invariances do not hold for all layers. For ResNet-18, we perform stitching at the first convolutional layer and each residual block, leading to 9 stitching locations. For VGG-16, we stitch at the first convolutional layer and all layers followed by max pooling operations, for a total of 6 stitching locations.

For ViT-Tiny, we follow Balogh & Jelasity (2025) and use a linear layer, applied to the feature representation of each token, to connect the front and end models. We perform stitching at the end of each of the 12 encoder blocks and at the final cls token before the classifier for a total of 13 stitching locations. For ViT-Base, we follow the same procedure used for ViT-Tiny, with the only exception being that we stitched at fewer encoder blocks (i.e., 6 blocks) totaling 7 stitching locations.

The stitching layer was trained for 30 and 12 epochs for the low- and high-resolution settings respectively, where we used the Adam (Kingma & Ba, 2015) optimizer with a learning rate of 0.001 and weight decay of $1 \times 10^{-4}$. Following Balogh & Jelasity (2023), we perform data augmentation (i.e., those used to train the standalone end models) for the standard model stitching, while for the invariance-aware model stitching we turned off data augmentation. Note that transformations applied to $\mathcal{D}_{\text{train}}^{\text{IRIs}}$ are not guaranteed to be within the identically represented inputs (IRIs) set, breaking the forward–backward compatibility operationalization. The alternative to conducting the dataset inversion on the augmented $\mathcal{D}_{\text{train}}^{\text{IRIs}}$ is impractical given its computational demands. For completeness, we note that controlling for augmentation (i.e., by turning off augmentation even for standard stitching) had minimal effect and resulted in similar trends, that is, the difference in augmentation does not drive the difference in behavior between standard and invariance-aware model stitching. For brevity and more direct comparison with prior work, we report only the results where standard stitching uses augmentation.

We conducted all experiments across three randomly initialized runs. For each configuration, we used the same end model instance, while we used a different front model for each run.

**Model Stitching under Adversarial Training:** When performing stitching under adversarial training, we consider two $\alpha$ configurations, $\alpha = 0$ (clean) and $\alpha = 1.0$ (adversarial) corresponding to optimizing the stitching alignment only on benign or adversarial samples respectively. The adversarial examples were generated with respect to the stitched composition using the same hyperparameters as those used to train the robust standalone models.

## C. Agreement Probability

Given a sample $x$, an end model $g : \mathcal{X} \to \mathcal{W}$ and a stitched composition $h : \mathcal{X} \to \mathcal{W}$, with $\mathcal{W}$ representing the space of post-softmax probability vectors, we compute the per-sample agreement probability as:

$$\text{agreement probability} = \frac{g(x)^T h(x)}{\max\left(g(x)^T g(x), h(x)^T h(x)\right)}. \tag{8}$$

The normalization term (i.e., the denominator) was introduced to avoid having confidently predicted samples dominate the agreement probability when averaging across the $\mathcal{D}_{\text{test}}$.

## D. Functional Latent Alignment (FuLA)

When implementing the FuLA objective in practice, we considered the Hints including and following the stitch level that overlap with the stitching location used for each architecture as defined in App. B. However, we note that other alternatives are also viable. Given our choice of stitching locations, when optimizing for FuLA, the stitching layer receives supervision signal from relatively distributed depth-levels ranging from low to high (with respect to the stitch level). We note that FuLA was defined such that no output-level signal is received, that is, only Hints up until the penultimate layer (i.e., the last feature map or cls token before the classification head for convolution- and transformer-based architectures respectively) were considered. In Fig. 6 we provide a visual overview of the different stitching alignment objectives and their relationship to FuLA.

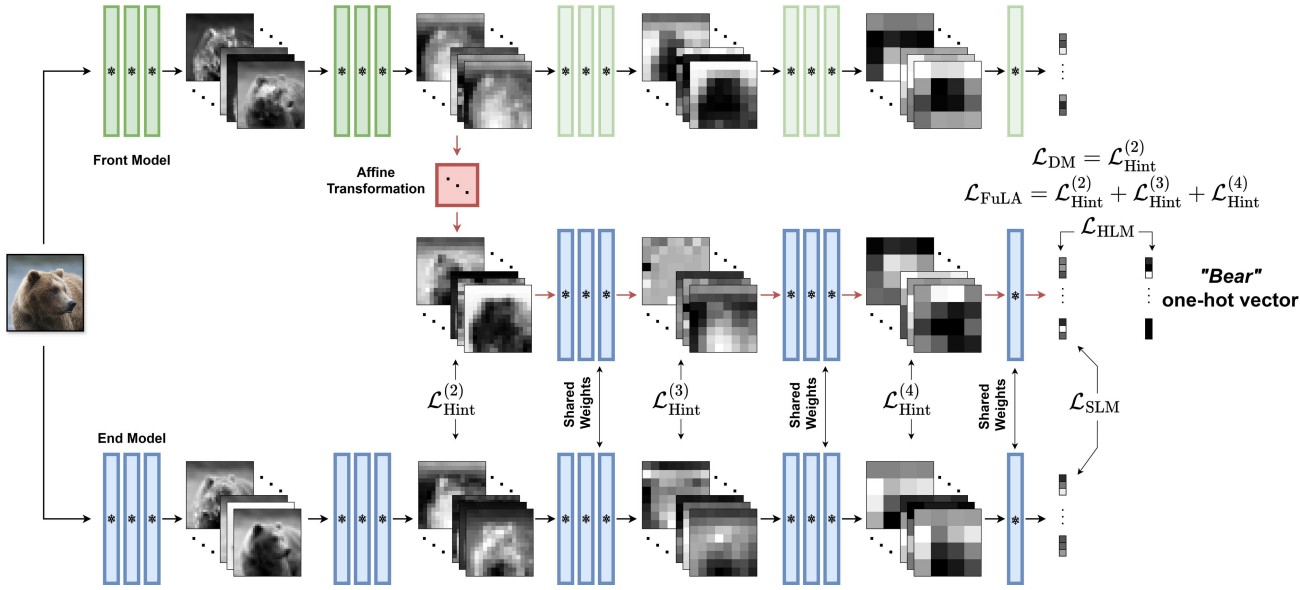

$$\mathcal{L}_{\mathrm{DM}} = \mathcal{L}_{\mathrm{Hint}}^{(2)}$$
$$\mathcal{L}_{\mathrm{FuLA}} = \mathcal{L}_{\mathrm{Hint}}^{(2)} + \mathcal{L}_{\mathrm{Hint}}^{(3)} + \mathcal{L}_{\mathrm{Hint}}^{(4)}$$

*Figure 6.* Overview of the different objectives in model stitching for functional similarity evaluation.

# E. Injecting Data with Shortcut Cues

We consider shortcut settings of varying availability, namely: (i) pixel-injected data at a fixed location for each class, (ii) pixel-injected data, or (iii) data from (ii) where the pixel patterns are replaced with random noise (i.e., negating the shortcuts) (see Fig. 7).

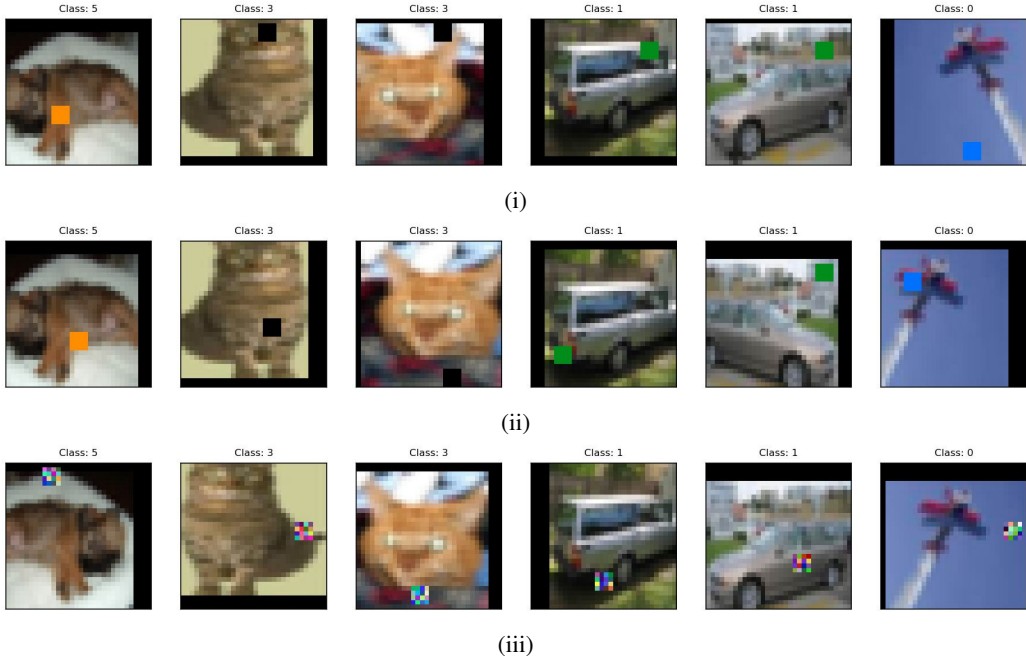

*Figure 7.* Pixel shortcuts of varying availability. Note that the shortcuts become less available as we transition from top to bottom row.

# F. Results on Additional Architectures

In this section we repeat the experiments conducted in the main paper on additional architectures (i.e., ResNet-18, VGG-16, ViT-Tiny and ViT-Base) while including results on data of higher resolution.

## F.1. IRIs Sanity Check

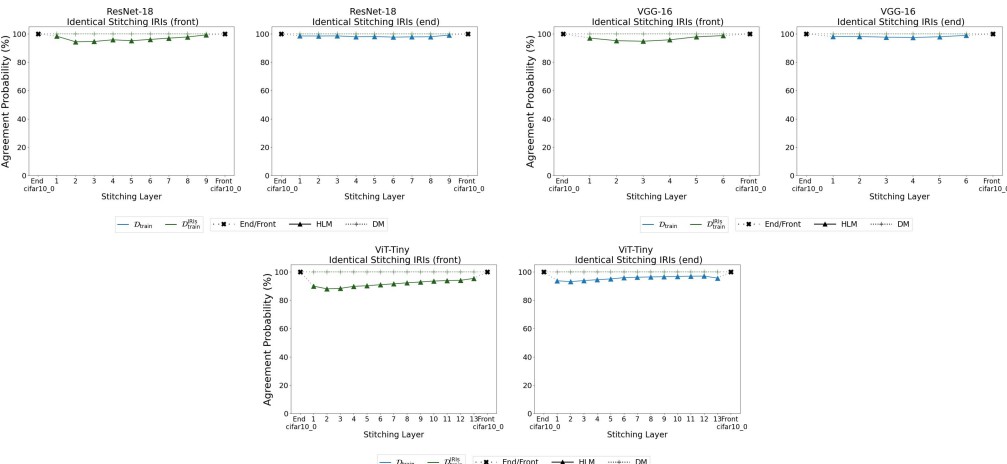

*Figure 8.* Stitching between identical models using corresponding pairs from $\mathcal{D}_{\text{train}}$ and $\mathcal{D}_{\text{train}}^{\text{IRIs}}$ (CIFAR-10).

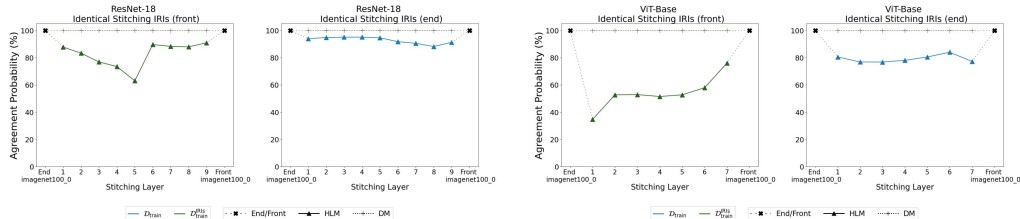

*Figure 9.* Stitching between identical models using corresponding pairs from $\mathcal{D}_{\text{train}}$ and $\mathcal{D}_{\text{train}}^{\text{IRIs}}$ (ImageNet-100).

To ensure that the $\mathcal{D}_{\text{train}}^{\text{IRIs}}$ resembles the original data in $\mathcal{D}_{\text{train}}$ with respect to the front model, at the stitch level, we perform model stitching between identical models (i.e., identical stitching) where the front model was given the original input in $\mathcal{D}_{\text{train}}$ and the end model was given the corresponding sample from the $\mathcal{D}_{\text{train}}^{\text{IRIs}}$ and vice versa. Given that the front and end models are identical, any deviation from the optimal agreement can be attributed to the data being perceived differently by the same model. In Fig. 8 we observe that in all cases the stitching alignment is close to perfect for the direct matching (DM) objective indicating that the $\mathcal{D}_{\text{train}}^{\text{IRIs}}$ and $\mathcal{D}_{\text{train}}$ are almost identical at the stitch level allowing us to probe for forward–backward compatibility in accordance with the forward–backward compatibility requirement defined in the main paper.

Hard label matching (HLM) also achieving relatively high agreement between the two configurations indicates that the relaxed IRIs preserve the semantic structure of the regular data. For CIFAR-10, we observe minimal divergence, whereas for ImageNet-100 we observe noticeable divergence either by the early or the mid layers. However, the functional similarity achieved under the invariance-aware setting was for the most part strictly decaying with respect to the layer's depth and therefore cannot be attributed to the relaxed IRIs drifting semantically.

### F.2. Functional Similarity across Information Variants

In Figs. 10 and 11 we observe similar trends across all architectures and datasets, where in regular model stitching the agreement probability is roughly interpolated between the end and front models. Although, for the cross-bias configuration, DM constitutes a common exception to this trend, high similarity is recovered eventually by the deeper layers for ResNet-18 and VGG-16 or by the last layer for ViT-Tiny and ViT-Base. On the other hand, under the invariance-aware model stitching, the drops in agreement are sharper in cases where we would intuitively expect lower similarity. Among the invariance-enabling formulations, FuLA and soft label matching (SLM) achieve higher similarity in convolution-based and transformer-based architectures respectively. However, we note that in the identity stitching configuration in ViT-Base, we observe that SLM fails to retrieve the optimal transformation despite it being trivial. We observe the effect both for ResNet-18 and ViT-Base when trained on ImageNet-100, however, the failure is significantly more pronounced for ViT-Base in which case we consider the sanity check to have failed (see Fig. 11, second row, first column).

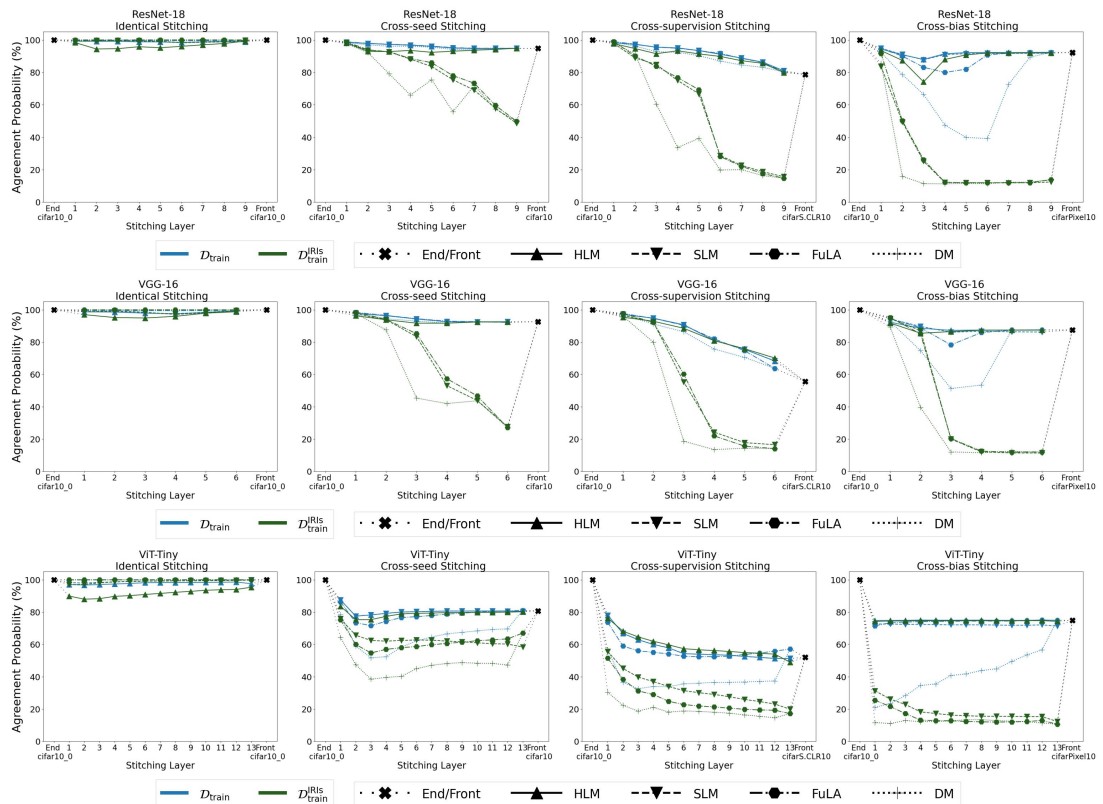

*Figure 10.* Stitching between models relying on different information (CIFAR-10).

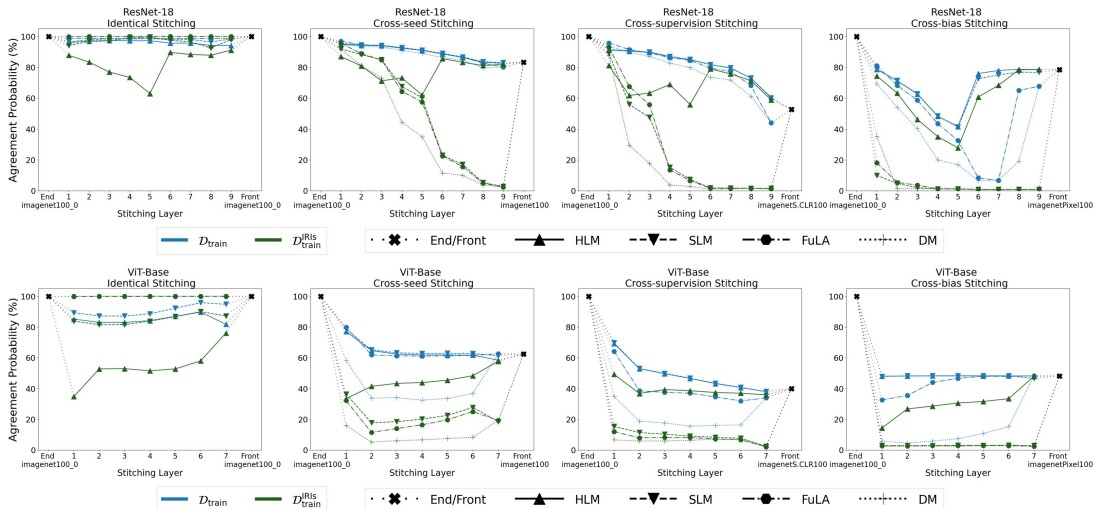

*Figure 11.* Stitching between models relying on different information (ImageNet-100).

### F.3. Functional Similarity under Unseen Predictive Information

Figs. 12 and 13 suggest similar trends across all architectures, where under regular model stitching all settings are susceptible to relying on shortcuts when establishing the stitching alignment. In contrast, invariance-aware model stitching is resistant to exploiting these shortcuts. We observe that under invariance-aware model stitching, the SLM formulation still engaged in shortcut learning but to a significantly lesser extent compared to the standard model stitching. For convolution-based architectures and the ViT-Base, the invariance-aware stitching under FuLA and DM is comparable in terms of largely mitigating the unseen cue exploitation, while for ViT-Tiny DM is significantly more effective. Note that HLM, even when trained on the IRIs data, still behaves similarly to the regular methods as it does not incorporate supervision signal from the standalone end model (i.e., HLM not being an invariance-enabling setting).

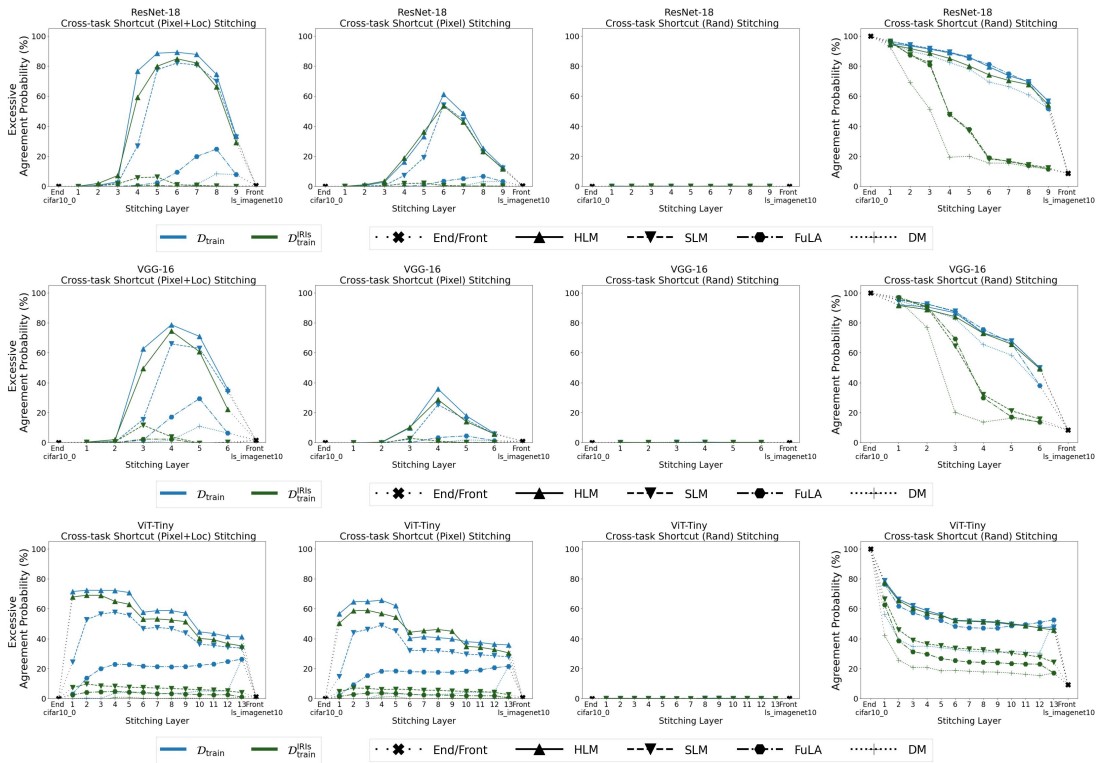

*Figure 12.* Cross-task stitching under shortcuts of varying availability (CIFAR-10).

We repeated part of the analysis using models trained on the same task (i.e., cross-seed stitching) and found that invariance-aware stitching mitigates the unseen cue exploitation behavior even in cases where the effect is significantly less pronounced to begin with, as shown in Fig. 14.

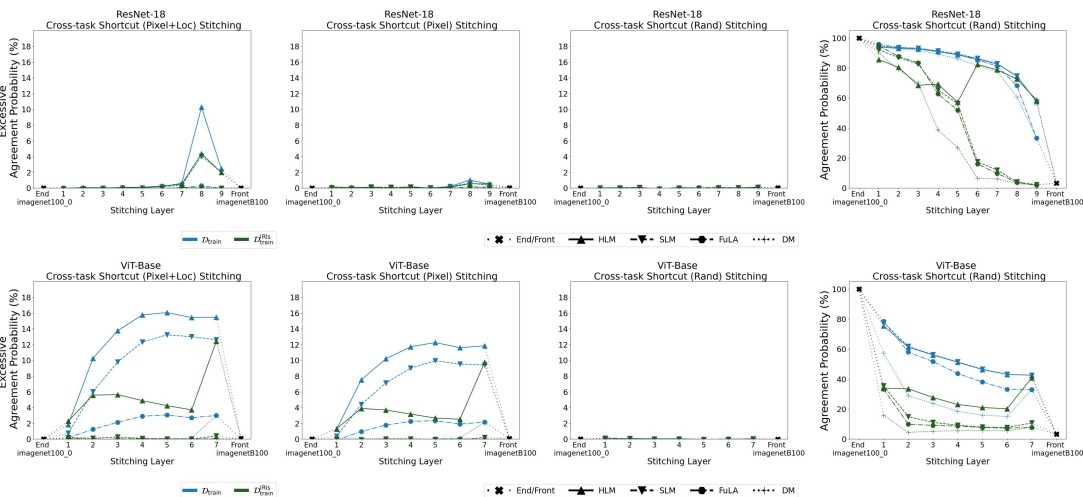

*Figure 13.* Cross-task stitching under shortcuts of varying availability (ImageNet-100).

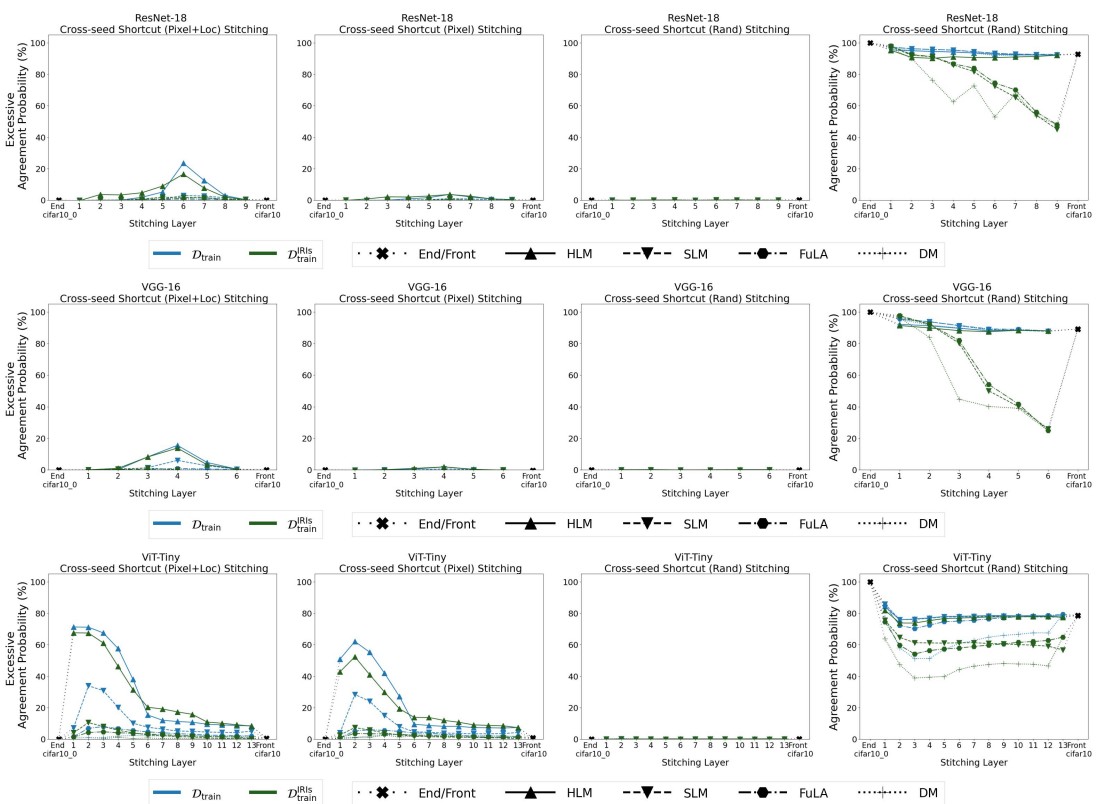

*Figure 14.* Cross-seed stitching under shortcuts of varying availability (CIFAR-10).

## F.4. Functional Similarity between Robust and Non-Robust Models

In Fig. 15, we provide results on robust-to-non-robust model stitching configurations. The analysis carried out on ResNet-18 generalizes to VGG-16 too. First, we observe that invariance-aware model stitching consistently reveals that robust and non-robust models are not as functionally similar as standard model stitching suggests.

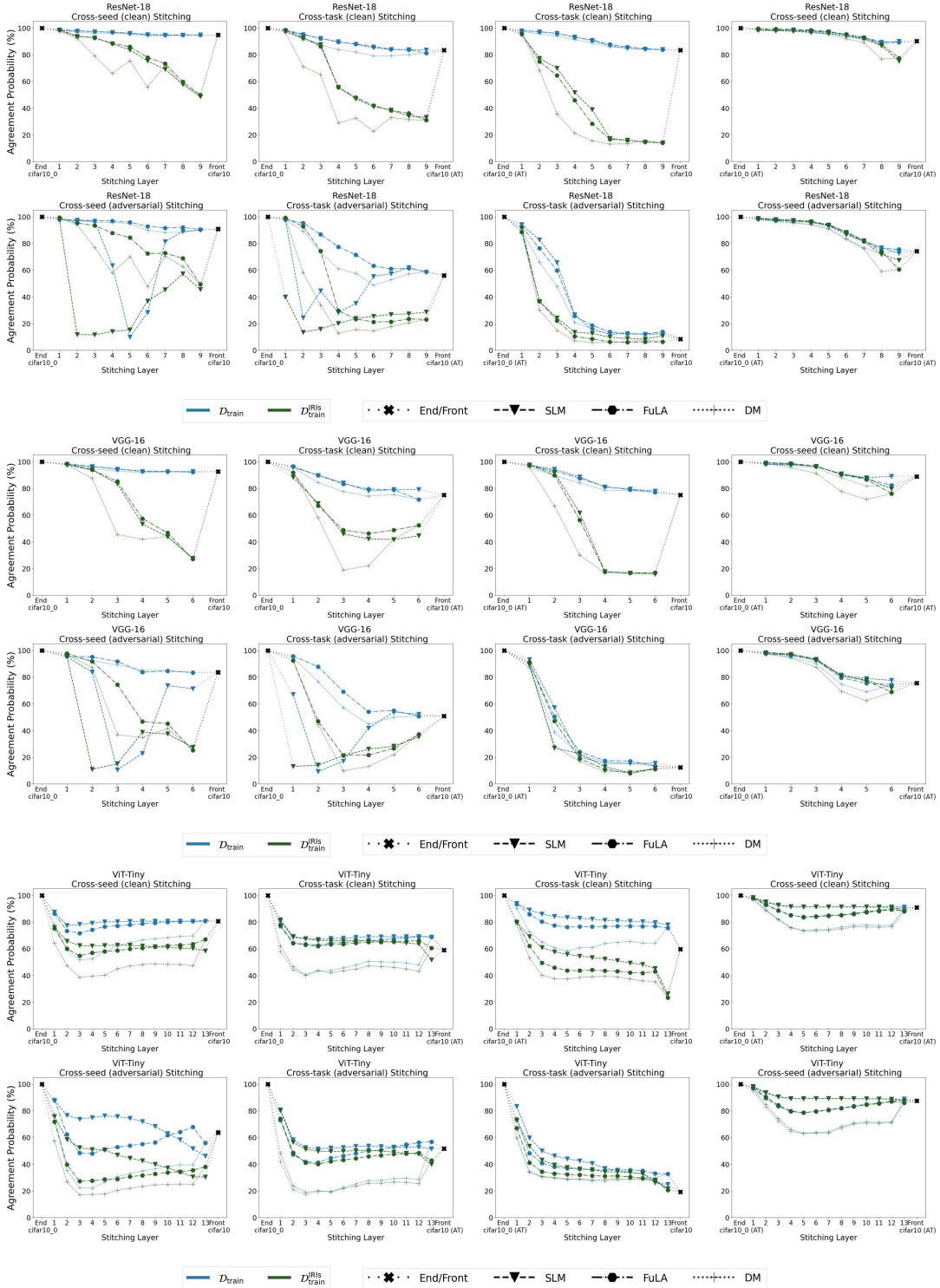

*Figure 15.* Stitching between robust and non-robust models (CIFAR-10).

Additionally, under the invariance-aware model stitching, FuLA generally achieves higher similarity than DM while remaining stable across all configurations. In contrast, SLM displays erratic behavior in configurations involving non-robust end models where the stitching alignment was established exclusively on adversarial examples. Again, under invariance-aware model stitching, robust models display more apparent functional convergence as robust-to-robust configurations achieve the highest overall agreement compared to all other configurations, a property missed by standard stitching.

For ViT-Tiny, we again observe that invariance-aware stitching reveals functional incompatibilities missed by standard model stitching. Moreover, standard model stitching fails to reveal that robust-to-robust configurations are significantly more functionally similar compared to other configurations. Unlike in convolution-based architectures, SLM behaves stably while generally achieving higher similarity compared to FuLA, rendering the former the overall better setting for ViT-Tiny.

Repeating the same analysis for ImageNet-100 on ResNet-18 and ViT-Base[5] as found in Fig. 16 leads to similar conclusions.

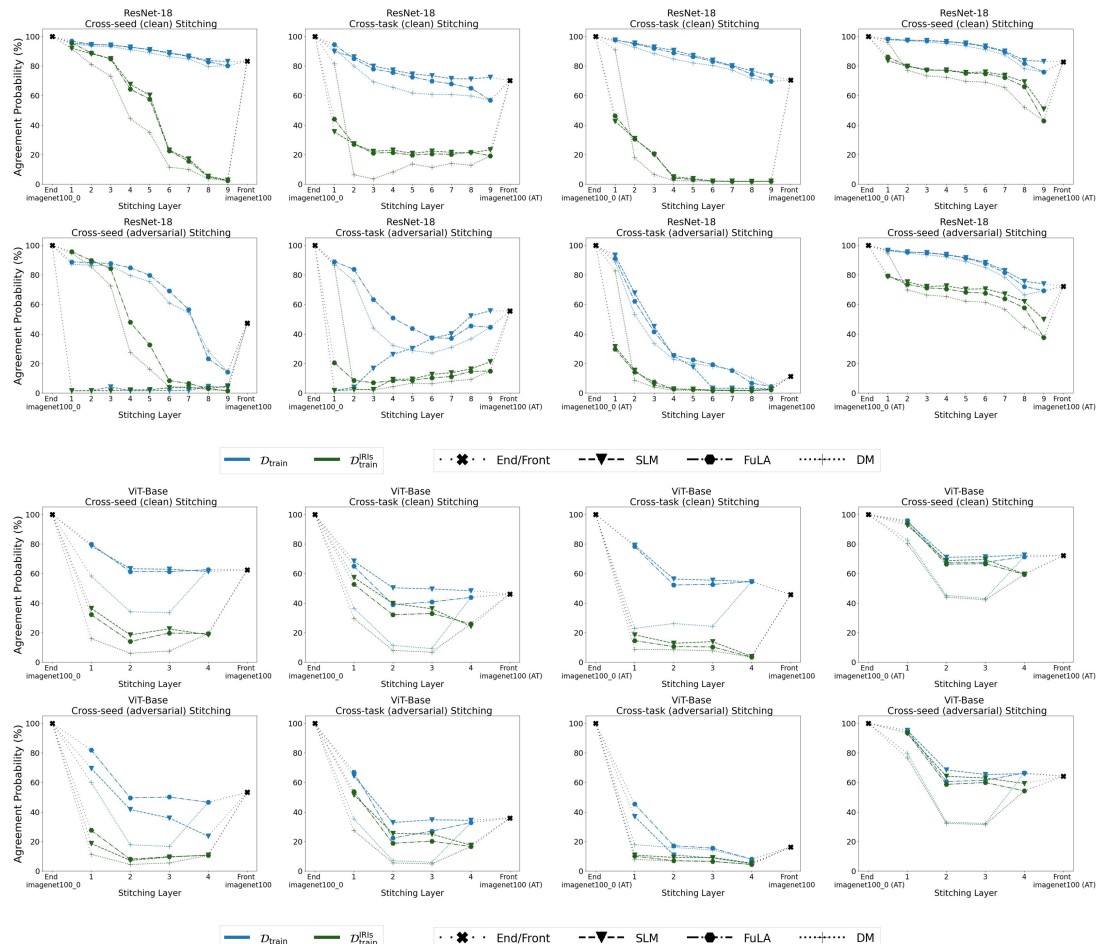

*Figure 16.* Stitching between robust and non-robust models (ImageNet-100).

# G. Relationship between FuLA and CKA

In this section, we investigate the relationship between a commonly used representation similarity metric, CKA (Kornblith et al., 2019), and functional similarity as evaluated by model stitching. For model stitching, we considered FuLA, as a representative formulation (i.e., being the middle ground between stitch-level and output-level formulations). We compare both the standard and the invariance-aware variants of these methods. The invariance-aware CKA was computed by first

---

[5]Given the large computational demand of generating adversarial examples for large datasets, the smooth transition trend between consecutive stitching locations and the fact that the results largely generalized across the previous architectures, we conducted the analysis for every other stitching location compared to the rest of the paper, that is, we stitched at 4 locations instead of 7.

generating the $\mathcal{D}_{\text{test}}^{\text{IRIs}}$ (i.e., IRIs (Nanda et al., 2022) of the regular $\mathcal{D}_{\text{test}}$ with respect to the front model) and in turn evaluating the standard CKA on that set. This is conceptually identical to the methodology of Jones et al. (2022) modulo the inversion methodology. For direct comparisons between these two metrics, we normalized the functional similarity such that it falls within $[0, 1]$.

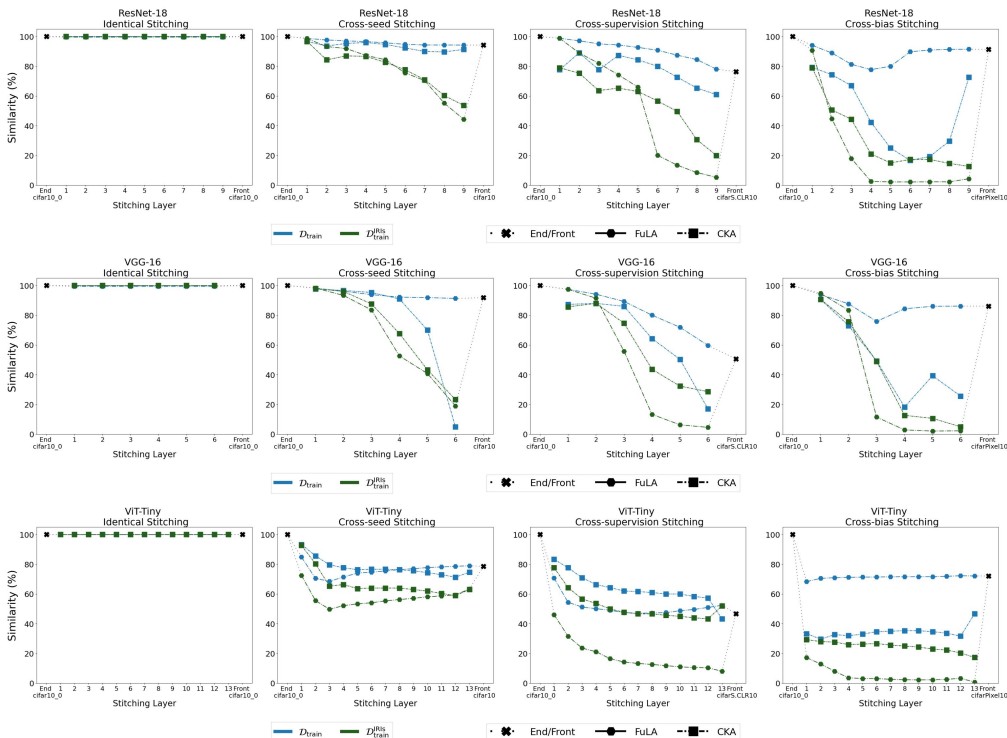

*Figure 17.* Comparing FuLA and CKA (CIFAR-10).

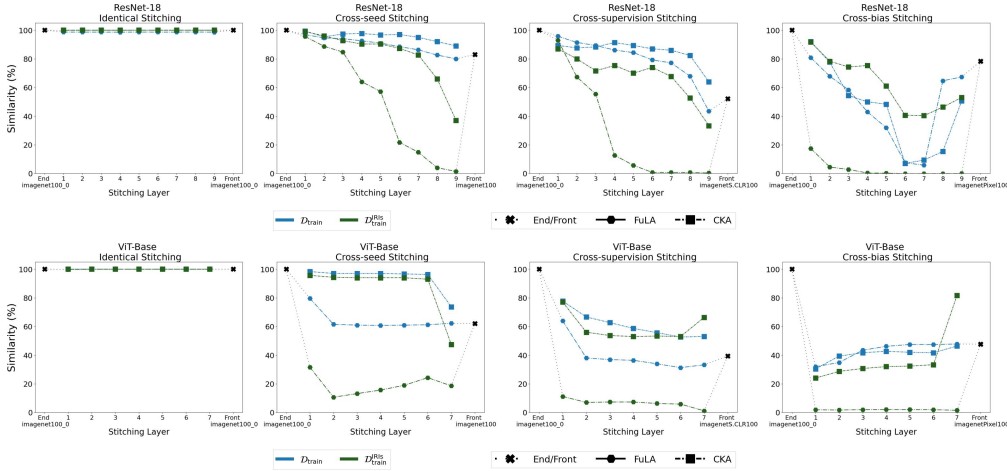

*Figure 18.* Comparing FuLA and CKA (ImageNet-100).

Compared to functional similarity, CKA is more difficult to interpret as its numerical values do not carry any intuitive meaning. Based on Figs. 17 and 18, we observe that the invariance-aware alternatives reveal discrepancies missed by their corresponding standard methods. Additionally, invariance-aware CKA can deviate significantly from the functional similarity inferred by invariance-aware model stitching, suggesting that unique insights are drawn by conducting invariance-aware model stitching.

