# OpenReview forum: "Grounding Functional Similarity by Invariance-Aware Model Stitching"
_ICML.cc/2026/Conference — ICML 2026 regular_

### Official Review · Reviewer_PZzv · 2026-02-20

**Soundness:** 3
**Presentation:** 2
**Significance:** 3
**Originality:** 3
**Overall Recommendation:** 5
**Confidence:** 4

**Summary:**

The paper proposes an explanation for the previously observed phenomenon that typical model stitching lets stitched models appear similar although they rely on different features. A new stitching method is proposed that is claimed to give more meaningful similarity estimates.

In regular stitching, models are functionally similar if the outputs of the stitched model are similar to the outputs of the original models, that is the representations of the front model (modulo stitching transformation) are compatible with the end model (“forward compatibility”). The new stitching method adds requirements to consider models functionally similar: inputs that are represented identically by the front model (i.e., the model is invariant to the changes between them) must be processed similarly by the stitched model AND the end model by itself (“forward-backward compatibility”, the backwards comes from the sharing of input invariances by both models).

This new approach is operationalized with synthetically generated data that leads to almost identical representations at the stitch layer. Stitching with the new approach is compared to different formulations of the regular stitching objective, using image classification networks. The authors find that a) regular stitching is influenced by new input shortcut features that were not seen during training of the models to be stitched, b) the new stitching approach mitigates both the issue from prior work and a), and c) use this approach to revisit prior results on similarity of adversarially robust models, finding  that robust and non-robust models are less compatible under the new compatibility requirement compared to prior work with the forward-compatibility requirement.

Altogether, they find a new problem with regular stitching, propose an improved stitching approach, and demonstrate how it enables new observations.

**Compliance With Llm Reviewing Policy:**

Affirmed.

**Final Justification:**

My main concerns were regarding presentation, which were addressed in the rebuttal and can reasonably be fixed for a camera-ready version. While they dropped a claim during the rebuttal phase, I still believe that the paper has a meaningful contribution.

**Key Questions For Authors:**

* Q1: How large is $D_{train}^{IRIs}$ compared to $D_{train}$? As I understand (L183), you replace each sample with exactly one sample of its equivalence class (via IRI). Then you train the stitching layer with $D_{train}^{IRIs}$ ignoring the original training set from now on. But you define forward-backward compatibility via _pairs of inputs that are represented identically_. The training data does not have such pairs, so I do not understand how you test the requirement with it. Instead, you only study aggregated behavior. Do I understand this correctly? If yes, I feel like this is a limitation in that you are not directly testing the requirement.
* Q2: Could you please explain once again the link between IRI data, forward, and backward compatibility? As I see it, regular stitching shows forward compatibility. Using IRIs without stitching and simply comparing outputs would give us information about shared invariances. Does stitching and IRIs really give us information about both concepts at the same time? I believe it works because you are comparing the stitched model to the end model, but laying this out clearly could really help understand your contribution.
* Q3: L326 col1: What does permuted across classes mean? Each class simply gets mapped to a differently colored shortcut pixel?
* Q4: There is currently no mention of a code release. Should it be uploaded to openreview, I cannot access it. What are your plans here?

If you can clear up some of my confusion and make code and data publicly available, I am willing to increase my score.

**Limitations:**

yes

**Strengths And Weaknesses:**

## Strengths
* Soundness: The claims are supported by solid evidence
* Soundness: The experiments reasonably isolate the important factors
* Presentation: I was able to follow the overall structure well
* Significance: Since there is typically no ground truth similarity, grounding results in intuition is important. Demonstrating the failure cases and proposing methods that align with intuition better is key to getting conclusions that are robust. This paper makes an important contribution towards more reliable similarity studies with goals in interpretability.
* Originality: Similar approaches to get more robust similarity estimates have been used in prior work [1, 2], but the combination of methods and application to model stitching is novel as far as I know.

## Weaknesses
- I think the clarity of the paper can be improved. I got confused multiple times throughout it. Below, a list of parts that were unclear to me and possible:
  - I got quite confused with term forward-backward compatibility initially. With backward, I would associate gradients flowing backwards through the net. I understand that this is unrelated to your requirement, but maybe changing the terminology could create a clearer picture for the readers. Since you have “invariant aware” in the title, “invariance-aware forward compatibility” may capture the concept better.
  - Sec 2.2. Paragraph 1: unclear what this means, especially the sentence regarding DM
  - The data setup in Sec 3.2 is a little unclear to me, see questions below.
  - L362: referring to model (1) and (2) makes it hard for the reader to understand. I would suggest describing the models again.
  - I am unsure how the IRI data relates to the evaluation (more in questions below)
- Missing information: Sec 3.4 makes several claims. The evidence for it is in the appendix but not referred to in the text. Please add references to the corresponding appendix sections.
- L33 col2: Wrong citation. Csiszárik et al. [3] were the first to distinguish the representational and functional similarity as far as I know. I think the citation of Ciernik should be replaced.
- Limited reproducibility: no code or data is provided.
- Missing related work: [1] propose a similar methodology to avoid similarity scores being inflated for models that rely on different cues. While [1] is cited in Sec 3.3, it is not acknowledged for its methodology.
- Minor presentation issues:
  - L89 col1: Sentence needs a period?
  - L151 col1: You use the DM acronym, which was given early in the introduction. Since there were quite a few acronyms, a reader might benefit from having the full words repeated here.
  - L184 col2: broken sentence.
  - L405 col1: “nun” -> “non”
  - L429 col1: “representative” occurs twice
  - Figure 7: the notation looks like the loss terms are squared, cubed, etc. Writing ^{(2)} instead of ^2 would eliminate some ambiguity.
  - L1108: missing word, probably “insights”

[1] Jones et al. 2022. If you’ve trained one you’ve trained them all: inter-architecture similarity increases with robustness.
[2] Nanda et al. 2022. Measuring Representational Robustness of Neural Networks Through Shared Invariances.
[3] Csiszárik et al. 2021. Similarity and Matching of Neural Network Representations.

---

> ### Author Rebuttal · Authors · 2026-03-31
>
> Dear PZzv,
>
> Thank you for taking the time to review our work.
>
> **Weaknesses:**
>
> We appreciate the careful reading as well as the suggestions and points brought up here. We will account for all of these in the updated version of the manuscript. Regarding the forward-backward terminology, instead of renaming it, we have included a note in L99/1 explicitly distinguishing the term backward from gradient backpropagation.
>
> &nbsp;
>
> **Questions:**
>
> > Q1a.  On training data and stitching.
>
> Correct, the $D^{IRIs}\_{train}$ mirrors $D\_{train}$ and thus has the same size. Your question revealed a subtle source of confusion on our end.
>
> - Training: When probing for the forward-backward compatibility, indeed the $D_{train}$ was dropped.
> - Initialization: Following standard practices L194-197/2, we used $500$ random samples from $D_{train}$ for initializing the stitching layer both when probing for forward and forward-backward compatibility.
>
> Csiszárik et al. suggest that stitching layer initialization influences the final alignment. We thus use identical initialization schemes to ensure that the observed difference between forward and forward-backward operationalizations is not an artifact of poor initialization. Accordingly, we have updated the L194-197/2 to document this choice.
>
> > Q1b. On the aggregated behavior.
>
> Correct, we do not directly probe for the forward-backward compatibility req. on a pairwise basis (as defined in L58-61/2). Instead we quantify the aggregate impact of its violation on a dataset basis (as described in L190-197/1). Our design combines the following:
>
> - minimal divergence from regular model stitching allowing us to mechanistically understand the difference in behavior.
> - sensitivity to inconsistencies in both forward and backward directions (kindly refer to Q2b).
>
> In model stitching for functional similarity evaluation, the interest is in characterizing the effect of interchanging representation, at the stitch-level, in aggregate. Based on that, we do not view our design as limiting in this context. However, we acknowledge that in applications where the interest is on characterizing per-sample behavior, alternative operationalizations are to be explored.
>
> &nbsp;
>
> > Q2a. Linking IRIs, forward/backward compatibility.
>
> Let us define analogous reqs. in isolation.
>
> Req. Forward compatibility: For each input, the standalone end model and the end model within the stitched composition process the stitch-level representation in a similar manner.
>
> Req. Backward compatibility (Nanda et al. in model stitching terminology): For each pair of inputs that the front model represents identically at the stitch-level, the standalone end model processes these inputs in a similar manner.
>
> Model stitching: Correct, regular model stitching informs us on the extent to which the front and end models have interchangeable representations (i.e., forward compatible). However, it does not inform us about shared-invariances.
>
> Shared invariances: Correct, measuring whether the standalone end model processes the regular inputs and their corresponding IRIs similarly, informs us about shared invariances. However, measuring shared-invariances does not inform us on the extent to which the front and end models are forward compatible.
>
> >Q2b. Probing for forward-backward compatibility.
>
> Under invariance-aware model stitching, we operationalize jointly the forward and backward (i.e., forward-backward) compatibility notions.
>
> Note that when establishing the alignment on $D^{IRIs}\_{train}$ there are two avenues for achieving suboptimal alignment on $D\_{test}$. Namely, either due to genuinely lacking forward compatibility or due to the standalone end model providing inconsistent supervision signal (by virtue of not sharing the invariances induced by the front model). Based on this, the invariance-aware model stitching is sensitive to both forward and backward directions. Finally, as you pointed out and stated by us in L70-74/2, the operationalization requires a supervision signal by the standalone end model (i.e., to meaningfully provide information on whether the invariances are shared).
>
> &nbsp;
>
> > Q3. On permuting across classes.
>
> Correct, permuting the patterns across classes produces a dataset variant where the same patterns appear but are associated with different classes. E.g., in Fig 6 ii the blue pixel is originally predictive of the plane class, and in the permuted variant, is predictive of car but not plane.
>
> &nbsp;
>
> > Q4. On code & data.
>
> Upon acceptance, we will release our full codebase. Throughout the paper we used openly available datasets and targeted manipulation of these (e.g., added pixel shortcuts,  IRIs etc.). Given our codebase and the openly available datasets one can reproduce our results. To further increase the accessibility of our work, we will share our models’ weights while we intend to share, at least for the low resolution datasets, their IRIs corresponding to these front models.

---

> > ### Author Rebuttal · Reviewer_PZzv · 2026-04-03
> >
> > Thank you for the clarifications. Given my concerns where mainly regarding presentation, I will raise my score. Some smaller weaknesses like references to prior work were not addressed, presumably due to the character limit. Please consider these issues in your revision as well.

---

> > > ### Author Response · Authors · 2026-04-04
> > >
> > > Dear PZzv,
> > >
> > > Thank you for engaging with our rebuttal and for your positive assessment.
> > >
> > > All weaknesses not addressed  in our earlier response will be corrected as per your recommendation.
> > >
> > > We would like to inform you that we identified we overclaimed regarding unseen cue exploitation (L77/2), as upon reexamination, Smith et al. already provided evidence of this kind in Appendix G.2. Accordingly, we have dropped this as a standalone contribution from our manuscript and attribute the discovery of both failure modes to Smith et al. For the full context, kindly refer to our response to aNLT's rebuttal acknowledgment.
> > >
> > > Our remaining contributions are unaffected by this correction.

---

### Official Review · Reviewer_3pc3 · 2026-02-24

**Soundness:** 2
**Presentation:** 2
**Significance:** 3
**Originality:** 3
**Overall Recommendation:** 4
**Confidence:** 4

**Summary:**

The paper presents an analysis and a new method in the area of representational alignment and model stitching. It begins by arguing that standard model stitching evaluates functional similarity poorly because it relies only on forward compatibility. Since models can use completely different information cues (a nice example is pixel shortcuts) but still align well, standard metrics prove highly misleading. The authors therefore propose an "invariance-aware" stitching method requiring what they call a forward-backward compatibility. They generate different inputs that are represented identically, to ensure the stitched model handles the front model's invariances correctly. At its heart, the paper proposes a essentially a more rigorous sanity check for deep net alignment.

**Compliance With Llm Reviewing Policy:**

Affirmed.

**Final Justification:**

The rebuttal did address my concerns, to reinforce my initial positive view of the paper.

During the rebuttal phase, with the help of the reviewers, the authors realized that one of their contributions had already been published elsewhere (admittedly a hard find hidden in an Appendix!). I think the paper stands well on its legs regardless.

**Key Questions For Authors:**

Could you mathematically formalize the "preimage domain" shading in Figure 1? It currently reads as a conflation between a continuous input interval and a strict set-theoretic pre-image. I'd need this clarified to feel 100% convinced about the foundational framing.

In conceptual terms, how does your main contribution meaningfully depart from Nanda et al.'s (2022) use of IRIs?

Given the heavy reliance on acronyms, would it be possible to simplify or reorganize their presentation to make the exposition easier to follow for readers?

**Limitations:**

Yes.

**Strengths And Weaknesses:**

On the positive side, the paper is tackling a fundamental problem: current deep net alignment metrics are too forgiving. I believe this space is definitely in need of further research, and the discovery that standard stitching exploits unseen predictive cues to artificially inflate alignment is a good finding. The paper backs these claims with a strong empirical foundation. The experimental part covers robust vs. nonrobust models across architectures, adding empirical weight to the overall conclusions.

At the same time, I found the paper to be an incredibly dense, acronym-heavy read (TLM, SLM, HLM, DM, FuLA, IRIs). The narrative is not fluent, making the core insights harder to digest than they should be. This makes it tough for me to delineate the exact boundaries of novel contributions against what is already established in prior work. Several observations feel known from the literature (e.g. spurious alignment findings), blurring the line with papers like Smith et al or Nanda et al. As a result, the results put forward by the paper do not feel game changing, being closer to an incremental advancement over existing methodology.

One point of concern for me is the theoretical framing, starting from the teaser figure which I had some difficulty wrapping my head around. The figure defines $f(x) = 2|x - 0.25|$ and shades the continuous interval [0, 0.25] as the "Preimage domain" (one more unnecessary definition in my opinion). A pre-image is defined relative to a specific subset of the codomain, not just an arbitrary contiguous block of the input space. To me, the figure is visibly confusing the concept of a pre-image set (the inputs mapping to shared intermediate representations) with a continuous portion of the domain. While the geometric intuition is not completely baseless, formally labeling that block as the preimage domain is mathematically imprecise and front-loads the paper with confusing terminology.

---

> ### Author Rebuttal · Authors · 2026-03-31
>
> Dear 3pc3,
>
> Thank you for the effort put in reviewing our work and for posing thoughtful questions. Since the weaknesses and questions largely overlap, we respond to them jointly below
>
> **Questions**
>
> >Q1. On formalizing the preimage domain.
>
> We acknowledge that not explicitly defining the preimage domain in mathematical terms can be confusing. Concretely, let  $f: \mathbb{R} \to \mathbb{R}$ and $\mathcal{X}_a \subseteq \mathbb{R}$ the accessible domain. Given the image  $\mathcal{I}_f = f(\mathcal{X}_a)$, the preimage domain is defined as  $\mathcal{X}_p = f^{-1}(\mathcal{I}_f) \setminus \mathcal{X}_a$.
>
> Consequently, the accessible $\mathcal{X}_a$ and the preimage $\mathcal{X}_p$ domains are not arbitrarily defined on the input domain but are connected through the function $f$. Accordingly, we will update Fig. 1 to formally define the accessible and preimage domains and explicitly communicate that the shading reflects their causal connection.
>
> &nbsp;
>
> > Q2a. Blurred contributions wrt. model stitching.
>
> Towards sharpening the boundaries of our findings relative  to previous works, we elaborate on our main contributions as laid out in L74-99/2.
>
> Smith et al. demonstrated that models relying on different information cues can be stitched successfully.
>
> - In our work we identified an additional failure mode showing that regular model stitching can exploit predictive cues, previously unseen by both models, to improve the stitching alignment.
>
> - we conceptualized the forward-backward compatibility notion which we operationalized under invariance-aware model stitching showing that it mitigates both the previous (Smith et al.) and the newly identified failure modes.
>
> - we demonstrated the relevance of invariance-aware model stitching by revisiting robust to non-robust functional similarity and yielding novel insights in relation to previous work (Balogh and Jelasit, et al. 2023).
>
> - we analyzed the interplay of forward and backward compatibility across key stitching alignment formulations showing (i) incorporating backward compatibility is relevant across all formulations that incorporate supervision signal by the standalone end model and (ii) the level of supervision used during alignment establishes a trade-off between underestimating and inflating functional similarity.
>
> Overall, in the context of functional similarity evaluation by model stitching, our work meaningfully expanded upon previous findings on failure modes (Smith et al.) by (a) identifying a new failure mode, (b) mitigating a common source of both previous and newly identified failure modes, and (c) showing that novel insights are attainable under our framing.
>
> > Q2b. On conceptual difference wrt. measuring shared-invariances (Nanda et al.).
>
> Let us define analogous reqs. in isolation to express the concepts related to our work in a common conceptual frame.
>
> Req. Forward compatibility: For each input, the standalone end model and the end model within the stitched composition process the stitch-level representation in a similar manner.
>
> Req. Backward compatibility (Nanda et al. in model stitching terminology): For each pair of inputs that the front model represents identically at the stitch-level, the standalone end model processes these inputs in a similar manner.
>
> Contrasting the backward compatibility req. with the forward-backward compatibility req. (as defined in L58-61/2) one can see that probing for backward (i.e., measuring shared invariances) and forward-backward compatibility are related but fundamentally different concepts.
>
> Similar to Nanda et al., our operationalization of forward-backward compatibility uses IRIs. However, when probing for backward compatibility, we get information on whether two models share invariances but not whether these models are forward compatible. On the other hand, probing for forward compatibility informs us on whether the two models have interchangeable representations (i.e., forward compatible) but not about whether these models share invariances. Our forward-backward compatibility captures these two notions jointly providing information on (in)compatibility in both forward and backward directions simultaneously. For example, two models can share invariances but not be forward compatible, be forward compatible but not share invariances and anything in between.
>
> &nbsp;
>
> > Q3. On using many acronyms and dense writing.
>
> Our plan on improving upon these aspects is:
>
> - Provide a more detailed roadmap of Sec 2. with pointers to relevant subsections.
> - Repeat full acronym definitions upon first mention within each section.
> - Drop the TLM acronym.
> - Group all stitching objectives into a single subsection.
>
> We are also open to any additional suggestions by the reviewers.

---

> > ### Author Rebuttal · Reviewer_3pc3 · 2026-04-02
> >
> > The rebuttal addressed my points and resolved my observations about the mathematical framing.
> >
> > In the revision, please state the distinction from Nanda et al. in a very direct way early in the paper, ideally separating the shared invariances-only from joint fwd-bwd compatibility in one concise paragraph.

---

> > > ### Author Response · Authors · 2026-04-04
> > >
> > > Dear 3pc3,
> > >
> > > Thank you for going through our rebuttal and for your positive assessment.
> > >
> > > We would like to inform you that we identified we overclaimed regarding unseen cue exploitation (L77/2) as upon reexamination Smith et al. already provided evidence of this kind in Appendix G.2. Accordingly, we have dropped this as a standalone contribution from our manuscript and attribute the discovery of both failure modes to Smith et al. For the full context, kindly refer to our response to aNLT's rebuttal acknowledgment.
> > >
> > > Our remaining contributions are unaffected by this correction.

---

### Official Review · Reviewer_oB7g · 2026-03-05

**Soundness:** 3
**Presentation:** 3
**Significance:** 3
**Originality:** 3
**Overall Recommendation:** 5
**Confidence:** 4

**Summary:**

In this article, they consider measuring model similarity using model stitching.

They point out even if models can be stitched together to solve a task with high accuracy, the models might be relying on different cues
for their predictions and therefore they argue that such a measure is not sufficient to call models similar.
They propose a new objective for model stichting (FuLA) which encourages the stitched model to closer match the internal representations of the end model from the stitcing layer onwards.
They show in experiments on vision models that measuring agreement probability of models stitched with FuLA or using soft label matching, better match a desirable notion of similarity between models than when using hard label matching (HLM) or direct matching.

**Compliance With Llm Reviewing Policy:**

Affirmed.

**Final Justification:**

After the rebuttal, I continue to find the main claims well-supported that model stitching with HLM does not result in a desirable measure of similarity, while invariance aware model stitching is better.

I therefore maintain my recommendation to accept the paper for publication.

**Key Questions For Authors:**

The most important questions are **W1** and **Q1**.

**Limitations:**

yes

**Strengths And Weaknesses:**

**Strengths:**

**S1:** The article makes a good case for why we would not like to use model stitching with HLM as a measure of similarity between models.
For example, figure 2 shows that models using different cues are still highly similar if we consider their agreement probability after using HLM.


**S2:** The writing is clear and well-structured.




**Weaknesses:**

**W1:** The article would benefit from more discussion on why even FuLA and SLM give quite high similarity scores when the stitching layer is trained on the "clean" data $D_{train}$.
Even when the stitching is trained on $D_{train}$ should we not ideally be able to see a difference between models which use different cues?

**W2:** The experiments are limited to three types of vision classification models and only three random seeds are used for each measurement.


**Questions and suggestions:**

**Q1:** It doesn't seem like there is a big difference between using FuLA and soft label matching (SLM). Is there a good reason for using FuLA, even though SLM seems simpler?

**Q2:** Page 2, column 2, lines 94-96: "backward compatibility is applicable to all invariance-enabling formulations, yielding qualitatively analogous conclusions" I don't understand what you mean by this sentence.

**Q3:** Section 2.2: You should be more clear in this section that this is you new proposed method. This might be made even more clear if you add it to your contributions on page 2.

**Q4:** Please make sure to reference the relevant sections of the appendix. For example, when you mention agreement probability on page 5, you should reference appendix E with the precise definition.
Also, on page 8, make sure to reference the appendices where results from the experiments you mention can be found.

**Q5:** Typos: page 8, column 1, line 407: "extend" -> "extent". page 8, column 1, line 425: "evidence" -> "experiments".

---

> ### Author Rebuttal · Authors · 2026-03-31
>
> Dear oB7g,
>
> Thank you for taking the time to review our paper and for the constructive feedback.
>
> **Weaknesses:**
>
> >W1. On the general failure of regular model stitching under FuLA and SLM.
>
> Indeed, as shown in Fig. 2, regular model stitching (i.e., trained on $D_{train}$) under neither FuLA nor SLM is able to address the failure mode identified by Smith et al. under Hard Label Matching (HLM), namely that models trained on different cues can be stitched successfully. In fact, despite being limited to only stitch-level supervision, even Direct Matching (DM) is subject to this counterintuitive behavior by the deep layers. Based on this, the behavior appears to be fundamental to probing exclusively for forward compatibility (i.e., whether the stitch-level representations are interchangeable wrt. some alignment objective), as also argued in L246-249/2.
>
> We hypothesize that this is because, on regular data ($D_{train}$), the deep latent-level (e.g., DM performed at the late layers or FuLA) and output-level representations of the standalone end model encode sufficient task-level information to steer the stitching alignment towards behaving similarly to model stitching under HLM. Under this premise, it is not surprising that all forward compatibility objectives behave similarly, as the depth increases.
>
> &nbsp;
>
> > W2. On the limited number of experimental types and seeds.
>
> We would like to clarify that our experiments span four architectures (ResNet-18, VGG-16, ViT-Tiny and ViT-Base) across multiple datasets (CIFAR-10, LS-ImageNet-10 and ImageNet-100), while we followed standard practices using 3 seeds (Balogh & Jelasity, 2023). Although experiments beyond vision and classification are of relevance, we believe that our experimental scope sufficiently demonstrates the relevance of invariance-aware model stitching for better grounded functional similarity evaluation, and we leave future extensions to other modalities and tasks as a promising direction for future work.
>
>
> &nbsp;
>
> **Questions:**
>
> > Q1. Are there any benefits to using FuLA over simpler SLM?
>
> As also discussed in Sec. 5, an inherent limitation of our work is that we do not have access to an “oracle” notion of functional similarity, hence we can not be certain which formulation is optimal. Based on this, we resort to intuition-based arguments when deciding on which setting is optimal.
>
>
> In Secs 3.1 and 3.2 we have a good case on why invariance-aware model stitching is to be preferred over regular model stitching, however as discussed in Sec 3.4., similarly conclusive arguments could not be made when comparing across alignment objectives.
>
> Nevertheless, in the absence of any violations of grounded intuition, one should prefer the formulation that generally achieves the highest functional similarity (i.e., to avoid unnecessarily choosing suboptimal alignments). To this end we found that:
>
> - for convolutional architectures invariance-aware model stitching by FuLA appears to be the optimal choice, as it eliminates all failure modes while achieving stable behavior and in most cases, the highest functional similarity.
>
> - for transformer-based architectures, a trade-off emerged, namely, higher-level supervision (e.g., output-level vs latent-level) yielded higher functional similarity at the cost of increased susceptibility to failure modes.
>
> Based on these, FuLA under invariance-aware model stitching strikes a good balance between achieving non-trivial alignments and mitigating the failure modes of regular model stitching.
>
> &nbsp;
>
> > Q2. Backward compatibility is applicable to all invariance-enabling model stitching formulations, yielding qualitatively similar conclusions.
>
> By this we mean that when any invariance-enabling alignment objective (SLM, FuLA, or DM) is combined with backward compatibility, the resulting invariance-aware stitching consistently mitigates previous and newly identified failure modes of regular model stitching, while also revealing functional discrepancies previously obscured.
>
> &nbsp;
>
> >Q3. On highlighting FuLA as our contribution.
>
> We have updated Sec. 2.2 to make it explicitly clear that FuLA is a novel alignment method proposed by us.
>
> &nbsp;
>
> >Q4-5. On adding references to the Appendix and typos.
>
> Thank you for these pointers, we updated our manuscript accounting for these.

---

> > ### Author Rebuttal · Reviewer_oB7g · 2026-04-01
> >
> > The authors are not able to run a lot of extra experiments, but I also do not find this important for acceptance of the paper. It is only the reason why my recommendation to accept is not a strong accept.
> >
> > All in all, I will keep my recommendation to accept the paper.

---

> > > ### Author Response · Authors · 2026-04-04
> > >
> > > Dear oB7g,
> > >
> > > Thank you for reviewing our responses and for your positive assessment.
> > >
> > > We would like to inform you that we identified we overclaimed regarding unseen cue exploitation (L77/2) as upon reexamination Smith et al. already provided evidence of this kind in Appendix G.2. Accordingly, we have dropped this as a standalone contribution from our manuscript and attribute the discovery of both failure modes to Smith et al. For the full context, kindly refer to our response to aNLT's rebuttal acknowledgment.
> > >
> > > Our remaining contributions are unaffected by this correction.

---

### Official Review · Reviewer_aNLT · 2026-03-12

**Soundness:** 3
**Presentation:** 2
**Significance:** 3
**Originality:** 2
**Overall Recommendation:** 3
**Confidence:** 4

**Summary:**

The manuscript focuses on the question of how to measure functional similarity between neural networks using model stitching. The paper argues that standard stitching objectives can overestimate similarity because they only test forward compatibility and can miss differences in the invariances learned by the stitched models. The authors investigate whether stitching can be made more sensitive to invariance structure by incorporating backward compatibility. To do this, they propose an “invariance-aware” stitching setup that uses identically represented inputs (or metamers) during alignment, and they evaluate this across several stitching objectives. Empirically, they show that standard stitching can report high similarity even when models rely on different cues or exploit unseen shortcuts, whereas the invariance-aware setting often lowers similarity and reveals discrepancies (eg between robust and non-robust models).

**Compliance With Llm Reviewing Policy:**

Affirmed.

**Final Justification:**

Thank you for the rebuttal and the additional clarification. I appreciate the authors’ effort to refine some of the claims. However, I am still not persuaded that the paper’s main contribution is sufficiently novel relative to existing work that already shows stitching can overestimate similarity. In my view, the current paper provides a useful extension, but not yet a strong enough conceptual or methodological advance for ICML.

**Key Questions For Authors:**

Can the authors state more explicitly what the main contribution is? For example, is the contribution primarily conceptual, methodological, or empirical? It would also help to define more clearly which formulations count as “invariance-aware model stitching”; my understanding is that these are the variants trained on IRIs, but this should be stated very explicitly. Finally, how do the authors see this work as going beyond prior findings that stitching compatibility can be high despite different cues or different internal representations?

**Limitations:**

A key limitation is that the conclusions rely on relatively small-scale image-classification settings, mostly on CIFAR-10 and LS-ImageNet-style data, so external validity is uncertain. Another limitation is that the interpretation hinges on IRIs as probes of invariance structure, but most of these would be unnatural inputs (since they are synthesized form random noise), which complicates the interpretation. It is arguably more interesting to characterize invariances with respect to naturalistic variations. More generally, the paper does not provide a clear oracle notion of when two models should count as functionally similar, so it remains difficult to assess whether the proposed criterion is the right one or simply a stricter one.

**Strengths And Weaknesses:**

The paper addresses an important question: when does stitching-based compatibility reflect genuinely similar computation rather than merely task-level interchangeability? The distinction between forward compatibility and invariance-related compatibility is interesting, and the use of metamers to probe this is potentially useful. I also appreciated the effort to compare multiple stitching objectives rather than focusing on a single formulation and the robust vs nonrobust models is an interesting test case.

Weakness: The main issue is clarity. Several core concepts are hard to parse, including the definitions of forward compatibility, backward compatibility, and the exact meaning of “invariance-aware” stitching. Terms such as TLM, DM, and FuLA are introduced in a way that can feel jargon-heavy, and some sentences are difficult to follow. It is also not always clear what the main methodological contribution is: is it the forward-backward compatibility requirement, the IRI training setup, FuLA, or the empirical comparison among these? Relatedly, the novelty relative to recent work is not fully convincing. The paper repeatedly cites Smith et al. (2025), which already showed that high stitching alignment can be misleading, so it would help to state much more sharply what is new here. More broadly, some of the claims about “invariance structure” feel underspecified, and the paper does not engage enough with prior work on IRIs/metamers showing limited cross-model generalization (Feather et al 2023) or other work exploring the discrepancy between representational and functional similarity measures (Bo et al., 2025; Klabunde et al, 2023) . Finally, the experimental setting is somewhat narrow: most results are on relatively small models and datasets, so it is unclear how directly the conclusions extend to more mainstream modern models.

Relevant papers:
(1) https://arxiv.org/abs/2305.06329
(2) https://www.nature.com/articles/s41593-023-01442-0
(3) https://arxiv.org/abs/2411.14633

---

> ### Author Rebuttal · Authors · 2026-03-31
>
> Dear aNLT,
>
> Thank you for reviewing our work and for the detailed feedback.
>
> **Weaknesses:**
>
> > W1. Clarity and acronyms.
>
> We will improve upon this as:
>
> - Provide a more detailed roadmap of Sec 2. with pointers to relevant subsections.
> - Repeat full acronym definitions upon first mention within each section.
> - Drop the TLM acronym.
> - Group all stitching objectives into a single subsection.
>
> We are open to additional suggestions.
>
> &nbsp;
>
> > W2/Q1. Contributions.
>
> Our contributions are both conceptual and empirical.
>
> - Conceptual: we proposed the concept of forward-backward compatibility. To disambiguate this in relation to previously explored concepts we will map both forward (model stitching) and backward compatibility (measuring shared-invariances) in a common conceptual frame using stitching terminology.
>
> Req. Forward compatibility: For each input, the standalone end model and the end model within the stitched composition process the stitch-level representation in a similar manner.
> Req. Backward compatibility (Nanda et al.): For each pair of inputs that the front model represents identically at the stitch-level, the standalone end model processes these inputs in a similar manner.
>
> Forward-backward compatibility (L58-61/2) is conceptually different from either forward or backward notions in isolation, measuring invariances does not inform on representation interchangeability, whereas measuring forward compatibility does not inform on shared invariances. Our operationalization captures  the interaction of both.
>
> - Empirical: we (i) identified an additional failure mode, namely exploiting previously unseen cues to establish stitching alignment (ii) showed that invariance-aware model stitching mitigates both previous (Smith et al.) and newly identified failure modes and (iii) demonstrated the relevance of invariance-aware model stitching by revisiting robust to non-robust functional similarity and yielding novel insights
>
> - IRIs (Nanda et al.): not our contribution, we used to realize the forward-backward compatibility req. in model stitching.
> - FuLA: a secondary contribution for understanding the effect of supervision level when establishing stitching alignment.
>
> &nbsp;
>
> > W3. Limited engagement with the literature.
>
> Thank you for these pointers. We have added:
>
> - L370/2: In line with this, Feather et al. (2023) showed that standard-trained models develop idiosyncratic invariances that decrease under adversarial training.
> - L426/2: Klabunde et al. (2023) advocate for combining representational and functional perspectives for comprehensive similarity analysis. Our work resonates with this view within the functional domain, highlighting the importance ..
>  -L47/2: Included Bo et al. (2025) as an additional reference to “agnostic to functional behavior”
>
> &nbsp;
>
> > W4/L1. On small scale and narrow experimental suite
>
> Our experiments span four architectures (ResNet-18, VGG-16, ViT-Tiny, ViT-Base) across multiple datasets including ImageNet-100 (Sec 3.4), where key findings generalize across architecture and scale. We consider this scope to sufficiently support our claims. We leave expanding to larger models and modalities as future work.
>
> &nbsp;
>
> **Questions and Limitations:**
>
> > Q2. Invariance-aware model stitching.
>
> Correct, invariance-aware model stitching is operationalized by training the stitching alignment on $D^{IRIs}_{train}$
> and can only be realized under invariance-enabling formulations (i.e., those incorporating supervision from the standalone end model), as discussed in L68-74/2. Per your recommendation, we will state this explicitly at the end of Sec. 2.3.1.
>
> &nbsp;
>
> > Q3. On how our work expands upon Smith et al.
>
> Smith et al. showed that two models relying on different but previously seen cues (either by means of the front or both front and end models) can be stitched successfully. We identify an additional failure mode, that is, model stitching can opportunistically exploit cues previously unseen by both models, further undermining the reliability of regular model stitching. Importantly, invariance-aware model stitching mitigates both failure modes while providing insights on their shared origin.
>
> &nbsp;
>
> > L2.On unnatural IRIs and lack of oracle..
>
> We also consider this to be an interesting avenue. However, it is noteworthy that neural networks perceive the world differently from humans (Subramanian et al., NeurIPS 2023). IRIs capture the native invariance structure of the models themselves, which we argue serves as a principled base in the context of functional similarity by model stitching.
>
> The oracle limitation applies broadly to the field. In the absence of an oracle, grounding conclusions in intuition is a reasonable alternative. Beyond intuition, invariance-aware stitching mitigates the exploitation of previously unseen cues, a desirable property regardless of what the oracle notion might be, as we want to evaluate genuine model similarity rather than spurious alignment.

---

> > ### Author Rebuttal · Reviewer_aNLT · 2026-04-02
> >
> > Thank you for the rebuttal. I appreciate the response, but I am still not fully convinced about the degree of novelty beyond prior work that already illustrates that stitching can overestimate similarity. As a result, I am maintaining my original assessment.

---

> > > ### Author Response · Authors · 2026-04-04
> > >
> > > Dear aNLT,
> > >
> > > Thank you for taking the time to go through our responses.
> > >
> > > Upon careful reexamination of our claims against Smith et al., prompted by your unresolved concern on novelty, we identified that we overclaimed regarding unseen cue exploitation (L77/2). Accordingly, we will drop that claim and properly credit the finding to Smith et al.
> > >
> > > &nbsp;
> > >
> > > > On exploiting previously unseen cues.
> > >
> > > Particularly, Smith et al. (Appendix G.2) used the Hard Label Matching (HLM) objective to successfully stitch (i) an untrained front model into an MNIST-trained end model using single-colored images predictive of the class (Color-Only) and (ii) front and end models both trained on Color-Only using MNIST images.
> > >
> > > While these cases provide sufficient evidence for the phenomenon, which ultimately leads us to dropping our claim on L77/2, our analysis conducted in Sec 3.2 provides a more systematic account by controlling for stitching objectives, shortcut accessibility, and conflicting cues (i.e., realized by permuting pixel patterns across classes), enabling our contribution in L91/2.
> > >
> > > &nbsp;
> > >
> > > > Contribution beyond previous work.
> > >
> > > Smith et al. diagnosed two key failure modes of functional similarity evaluation by model stitching. Namely, model stitching (i) can not distinguish between models relying on different cues to solve their task and (ii) can exploit previously unseen cues to improve alignment.
> > >
> > > Our central contribution is the conceptual framing of forward-backward compatibility and its operationalization under invariance-aware model stitching, which was empirically shown to mitigate the failure modes identified by Smith et al. (Secs 3.1 and 3.2). Based on this, invariance-aware model stitching emerges as a more principled approach to functional similarity evaluation, revealing previously obscured functional discrepancies (Sec. 3.3).
> > >
> > > Finally, our work shifts the narrative from model stitching being misleading (Smith et al.) to invariance-aware model stitching being a promising avenue for functional similarity evaluation, meaningfully extending beyond prior work focused on diagnosing failure modes.

---

### Decision · Program_Chairs · 2026-04-30

**Decision:**

Accept (regular)

**Comment:**

This paper studies functional similarity via model stitching and shows that standard stitching can report high similarity even when models rely on different cues. to this end, it proposes to modify the stitching procedure by training alignment also on inputs constructed to yield identical intermediate representations, enforcing consistent behavior in these cases and making similarity sensitive to shared invariance

Reviewers agreed that the paper tackles an important problem and that the empirical analysis is technically sound. The proposed framework consistently reveals failure modes of standard stitching and provides more informative similarity estimates (e.g., in settings with shortcut features and robust vs. non-robust models).

The main concerns were limited novelty relative to prior work (e.g., Smith et al., Nanda et al.) and issues in clarity (dense presentation, heavy use of acronyms, and unclear definitions). While these weaknesses remain, they do not undermine the core contribution: a practically useful refinement of stitching-based similarity through joint forward–backward compatibility. The contribution remains slightly incremental but non-trivial and supported by careful experiments.

After discussion, three reviewers supported acceptance, and the remaining reviewer indicated they are comfortable with acceptance.
Overall, the paper is technically solid and provides a useful refinement of functional similarity evaluation that is likely to be of interest to ICML community, in particular for the representation similarity and model merging subfields. Therefore I recommend acceptance.

In the revisions the authors should implement all edits discussed, in particular:
- Clearly state the main contribution early and distinguish it from prior work.
- Simplify notation and reduce acronyms, especially in Section 2.
- Improve clarity of definitions and references to appendix results.